# Functional redundancy and formin-isoform independent localization of tropomyosin paralogs in *Saccharomyces cerevisiae*

Anubhav Dhar[1☯], V.T. Bagyashree[1☯], Sudipta Biswas[2], Jayanti Kumari[1], Amruta Sridhara[1], Subodh B. Jeevan[1], Shashank Shekhar[2], Saravanan Palani[1]*

**1** Department of Biochemistry, Indian Institute of Science, Bengaluru, Karnataka, India, **2** Departments of Physics, Cell Biology and Biochemistry, Emory University, Atlanta, Georgia, United States of America

☯ equal contribution
* spalani@iisc.ac.in

## Abstract

Tropomyosin is an actin-binding protein (ABP) which protects actin filaments from cofilin-mediated disassembly. Distinct tropomyosin isoforms have long been hypothesized to differentially sort to subcellular actin networks and impart distinct functionalities. Nevertheless, a mechanistic understanding of the interplay between Tpm isoforms and their functional contributions to actin dynamics has been lacking. In this study, we present and characterize mNeonGreen-Tpm fusion proteins that exhibit good functionality in cells as a sole copy, surpassing limitations of existing probes and enabling real-time dynamic tracking of Tpm-actin filaments *in vivo*. Using these functional Tpm fusion proteins, we find that *S. cerevisiae* Tpm isoforms, Tpm1 and Tpm2, colocalize on actin cables and indiscriminately bind to actin filaments nucleated by either formin isoform - Bnr1 and Bni1 *in vivo*, in contrast to the long-held paradigm of Tpm-formin pairing. We show that cellular Tpm levels regulate endocytosis by affecting the balance between linear and branched actin networks in yeast cells. Finally, we discover that Tpm2 can protect and organize functional actin cables in the absence of Tpm1. Overall, our work supports a concentration-dependent and formin isoform independent model of Tpm isoform binding to F-actin and demonstrates for the first time, the functional redundancy of the paralog Tpm2 in actin cable maintenance in *S. cerevisiae*.

## Author summary

A yeast cell's internal structure relies on a "highway" network of actin filaments stabilized by a protective protein called tropomyosin (Tpm). In budding yeast, there are two types of Tpm, which were believed to have different functions. Our study challenges this long-held view. To study these proteins, we first benchmarked a recently developed method to visualize these proteins in living cells,

**Data availability statement:** All relevant data are within the manuscript and its Supporting Information files.

**Funding:** This work was supported by a Department of Biotechnology-Wellcome Trust India Alliance intermediate fellowship (IA/I/21/1/505633), SERB SRG grant (SRG/2021/001600) and an Indian Institute of Science (IISc) start-up grant awarded to S.P. S.S. is supported by NIH NIGMS grant R35GM143050. The funders had no role in study design, data collection and analysis, decision to publish, or preparation of the manuscript.

**Competing interests:** The authors have declared that no competing interests exist.

and then, using the technology, discovered that the two Tpm types work together to stabilize all actin highways in the cell. Strikingly, simply increasing the lesser abundant Tpm type allows it to fully substitute in the absence of the other Tpm, protecting and organizing a functional network of actin highways. This reveals a previously unappreciated shared capability, where the lesser abundant Tpm type can act as a safeguard, ensuring the cell's crucial internal transport system remains robust during distress.

## Introduction

Tropomyosins (Tpm) are a major class of actin-binding proteins (ABPs) present in fungi [1] and metazoans [2–4]. Tropomyosins are helical coiled-coil proteins that form head-to-tail dimers and co-polymerize with actin filaments [5–8]. Tpm was initially discovered as a component of muscle [9] where it regulates actomyosin contraction [10,11]. Later studies revealed that Tpm isoforms are also abundant in non-muscle cells [4,12–14] and even single-celled eukaryotic organisms like fungi [13,15,16]. The canonical role of non-muscle Tpm is to protect actin filaments from actin-severing proteins like cofilin [17–19] and thus regulate turnover of actin networks [20]. Tpms display a great diversity of isoforms across species with involvement in various cellular processes such as cell motility [21], cell division [13], organellar movement, secretory vesicle delivery, etc. [2–4,22]. Mammals have four tropomyosin genes which produce around 40 isoforms via alternate splicing [2,14,23]. Various studies have shown that Tpm isoforms sort extensively to distinct actin filament subpopulations and define diverse functionalities for these networks by controlling the interactions between F-actin and other actin-binding proteins such as myosin, cofilin, capping protein, etc. [2,4,24–29]. The molecular basis for Tpm isoform sorting remains an unresolved question with various mechanisms such as concentration dependence [30] or a formin-dependent targeting [31,32] having been observed. Structural analysis using cryoEM has now revealed that Tpm isoforms may exert differential functions via different modes of binding and controlling accessibility of ABPs like myosin and cofilin to F-actin [33,34]. How Tpm isoforms have been fine-tuned to perform distinct spatial sorting and functions during evolution remains an active area of research and their crosstalk during recruitment and maturation within the same actin structure is just beginning to be understood [21,35].

The eukaryotic model organism *Saccharomyces cerevisiae* has two tropomyosin isoforms encoded by two different genes - *tpm1* and *tpm2* [1,15,16]. Tpm1 (199 amino acids) is considered a duplicated paralog of Tpm2 (161 amino acids) with a 38 amino acid internal duplication [16]. Tpm1 is considered the major isoform and Δ*tpm1* cells display lethality at higher temperatures and a near complete loss of actin cables [15]. Cells compromised in actin severing factors such as Δ*aip1* or Δ*srv2* mutants show partial restoration of cables in Δ*tpm1* cells which suggests that Tpm1 is majorly involved in protection of the actin filaments from severing proteins of the CCA (Cofilin-Coronin-Aip1) complex *in vivo* [36,37]. Δ*tpm2* cells on the other hand

show no detectable changes in actin cable organization and cell growth [16,38]. Interestingly, deletion of both *tpm1* and *tpm2* is synthetic lethal [16], suggesting an overlap of some essential functions. The minor isoform Tpm2 is expressed at ~5 fold lower levels than Tpm1 and Tpm2 overexpression could not restore normal growth and actin cables in Δ*tpm1* cells in a previous study [16]. Thus, the *in vivo* functions of Tpm2 have remained unclear but studies have suggested a role for Tpm2 in negatively controlling retrograde actin cable flow (RACF) possibly via regulating actin filament binding of the type-II myosin Myo1, which is positive regulator of RACF [39,40]. Both Tpm1 and Tpm2 facilitate processive motion of the type-V myosin Myo2 on coated actin filaments *in vitro* [41], contributing to the anterograde flow of organelles and cargo to the growing bud [22,42,43]. Thus, while Tpm1 is believed to play a major role in maintaining the actin cable cytoskeleton and dependent functions in yeast, the functions of Tpm2 are believed to be majorly distinct from Tpm1 [16].

Both Tpm1 and Tpm2 are known to localize to formin-nucleated actin cables in yeast through immunolabelling data [22] and recent genetic studies have indicated a possible preference between Tpm isoforms and formin isoforms [38], but whether Tpm1 and Tpm2 localize to distinct actin filaments/cables in *S. cerevisiae* has remained unaddressed mainly due to absence of functional Tpm fluorescent fusions and live-cell imaging data for Tpm1 and Tpm2. Recent work from our lab made it possible to visualize the live dynamics of Tpm across species using mNeonGreen-Tpm (mNG-Tpm) fusion proteins [44] for the first time and revealed their localization to actin cables and the actomyosin ring during the cell cycle in budding yeast, fission yeast, and to several actin structures in mammalian cells. mNG-Tpm fusion proteins contain a flexible 40 amino-acid long linker which separates the N-terminal mNeonGreen from the Tpm protein, which possibly allows for normal head-to-tail interactions between two Tpm dimers that would otherwise be hindered due to the N-terminal fluorophore in the absence of the linker. However, the combined action and roles of Tpm1 and Tpm2 in actin cable stability *in vivo* are still not understood.

In this study, we assess the functionality of mNeonGreen-Tpm fusion proteins and show that mNG-Tpm fusion proteins are functional and restore viability as sole copies in cells lacking native Tpm1 and Tpm2. We also engineer -AS- dipeptide containing mNG-ASTpm constructs to account for lack of N-terminal acetylation and improve functionality *in vivo*. Using mNG-ASTpm fusion proteins, we find that Tpm1 and Tpm2 indiscriminately associate with actin filaments made by either formin Bnr1 and Bni1, unlike the differential association of acetylated and unacetylated forms of *S. pombe* Cdc8 to distinct formin-nucleated filament populations shown in *S. pombe* [31,45]. We report a novel function of Tpm2 and show that Tpm2 can compensate for loss of Tpm1 upon increased expression *in vivo.* In contrast to the long-held view, we find that increased Tpm2 expression can restore full length actin cables and actin-cable dependent functions such as vesicle targeting to bud and maintenance of normal mitochondrial morphology suggesting that Tpm2 can independently organize a functional actin cable network in *S. cerevisiae* in the absence of Tpm1. Lastly, we also report the role of Tpm1 and Tpm2 in maintaining linear-to-branched actin network homeostasis in yeast, revealing an intriguing cascading effect of Tpm on actin dynamics at sites of endocytosis. Overall, our findings support a concentration-dependent and formin-independent localization of Tpm isoforms to actin cables and unveil the functional redundancy between Tpm isoforms in *S. cerevisiae.*

## Results

### mNG-Tpm fusion proteins are functionally-active tagged Tropomyosins

Visualization of tropomyosin in live cells has been a major challenge because fluorescently-tagged tropomyosin fusions are not completely functional and thus, need careful interpretation for accurate insights about Tpm localization and dynamics *in vivo* [46]. Recently, we have shown that mNeonGreen-Tpm (mNG-Tpm) fusion proteins containing a 40 amino-acid linker decorate cellular actin networks and clearly report localization of distinct Tpm isoforms to actin structures in fungal and mammalian cells [44] (S1A and S1B Fig), but their functionality compared to native Tpm remained untested. To address whether mNG-Tpm fusion proteins are functional tagged Tpm fusions and can compensate for loss of native Tpm in cells, we first assessed complementation of cellular defects caused by deletion of *tpm1* gene by mNG-Tpm1 expressed under the native *tpm1* promoter. We found that mNG-Tpm1 expression could near completely restore

normal growth (S1C, S1E and S1F Fig) but only partially restore the length of actin cables in Δ*tpm1* cells which exhibit slower growth and absence of actin cables (S1D Fig). The growth rescue despite the presence of shorter actin cables suggests (i) mNG-Tpm1 fusion is partially functional as compared to the native Tpm1 protein at native expression levels, and (ii) partial length cables may be sufficient to majorly support normal growth via proper cargo targeting to the bud. One possible reason why mNG-Tpm1 expression could only restore partial length actin cables is the lack of N-terminal acetylation of mNG-Tpm by the Nat3-Mdm20 complex which is essential for normal Tpm-actin binding *in vivo* [47–49]. This could result in reduced binding affinities towards F-actin as compared to native Tpm. To account for the lack of N-terminal acetylation, we constructed mNG-ASTpm, a modified mNG-Tpm fusion protein which contains an -Ala-Ser- (AS) dipeptide before the starting methionine of Tpm. The addition of the AS dipeptide is routinely used to restore normal binding affinities to Tpm protein expressed and purified from *E. coli,* possibly due to the stabilizing effect of the AS dipeptide on the N-terminal alpha-helix in the absence of the N-terminal acetylation [25,38,50–52]. Live-cell imaging of cells expressing either mNG-ASTpm1 or mNG-ASTpm2 fusion proteins from an integrated single copy showed clear labelling of actin cables and the actomyosin ring similar to mNG-Tpm1 and mNG-Tpm2 (Figs 1A, S1G, and S1H and S1 and S2 Movies), suggesting that addition of the -AS- dipeptide did not change the localization dynamics of the two isoforms. In terms of signal-to-noise, mNG-ASTpm1 and mNG-Tpm1 displayed comparable SNR while mNG-ASTpm2 showed significantly higher SNR than mNG-Tpm2 (S2A and S2B Fig). mNG-ASTpm1 expression could near completely rescue the growth defects in Δ*tpm1* cells (S1C–F Fig). Strikingly, in contrast to mNG-Tpm1, mNG-ASTpm1 restored full-length actin cables in Δ*tpm1* cells when expressed at similar levels (Fig 1B and 1C**),** suggesting improved functionality in terms of actin cable stability due to -AS-addition. These results clearly indicate that mNG-ASTpm1 is a functional tagged tropomyosin with similar activity as native Tpm1.

Next, we assessed if mNG-Tpm fusion proteins could also rescue synthetic lethality of Tpm1 and Tpm2. To test this, we expressed mNG-Tpm1, mNG-ASTpm1, mNG-Tpm2, mNG-ASTpm2 under their native promoters in Δ*tpm1*Δ*tpm2* cells containing a copy of Tpm1 in a *ura3*-centromeric plasmid. Shuffling of these strains to 5'FOA-containing media revealed that all mNG-Tpm fusions rescued synthetic lethality of Δ*tpm1*Δ*tpm2* cells when expressed from a high-copy plasmid (*pRS425*$_{(H)}$) (S2C Fig) but only mNG-Tpm1 and mNG-ASTpm1 restored viability when expressed from single-copy integration plasmid (*pRS305*$_{(I)}$) (S2D Fig). This suggests that both mNG-Tpm1 and mNG-Tpm2 fusions are functional tagged Tpm fusions and Tpm1 tolerates the presence of the N-terminal tag better than Tpm2 as mNG-Tpm2 imparts viability only at high expression levels. Next, we tested whether actin cables restored by mNG-Tpm1 and mNG-ASTpm1 are functional and assessed mitochondrial morphology as a readout of function. Mitochondria are known to physically interact with actin cables and their inheritance to the growing bud is dependent on Myo2 (type-V myosin) – mediated transport [53,54]. In addition, mitochondrial morphology is severely affected in the absence of Tpm1 when actin cables are lost, and hence, can be used as one of the markers for actin cable function in *S. cerevisiae*. We found that mitochondrial fragment number is restored to wildtype levels by expression of both mNG-Tpm1 and mNG-ASTpm1 in Δ*tpm1* cells which show hyper-fragmented mitochondria [53,55] (Fig 1D and 1E), suggesting again that polarized actin cables but not full length cables are sufficient for restoring downstream actin cable-dependent functions such as cargo transport and mitochondrial morphology. These data also suggest that Tpm-mediated actin cable stability *in vivo* may be one of the factors determining actin cable length control with cell size in yeast [56,57].

Taken together, our results demonstrate that mNG-Tpm fusion proteins are functional, and addition of an acetylation-mimicking -AS- dipeptide can improve their functionality in vivo, thus, expanding their applicability to probe a broad range of questions about Tpm isoform biology.

## Tpm1 and Tpm2 indiscriminately bind to actin cables nucleated by both formin isoforms

Tpm isoforms are known to sort to distinct actin filament structures in space and time in various organisms [4,24,32,33,45,58–62] and the basis of this spatial sorting remains an enigmatic question [30,63]. N-terminal acetylated

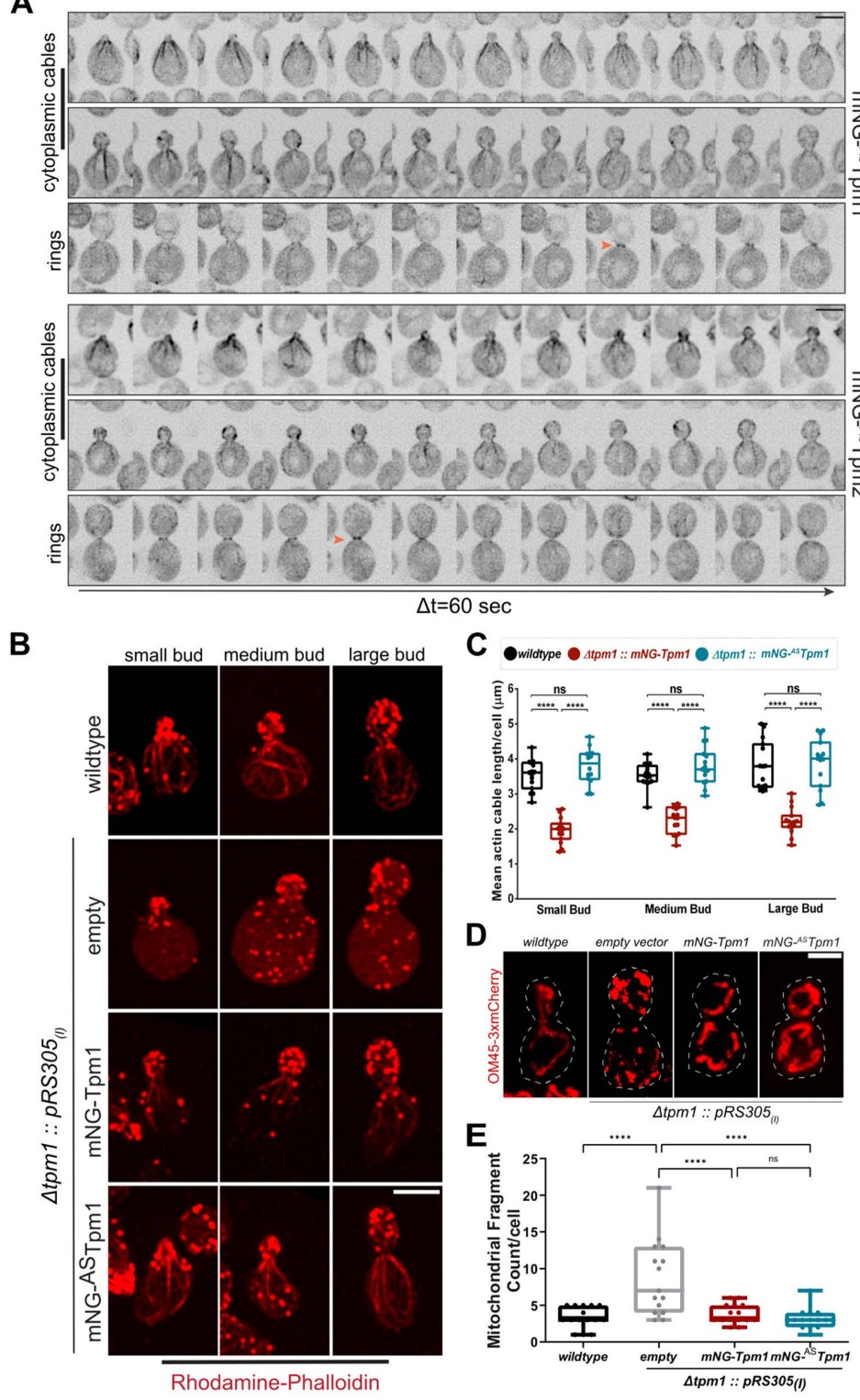

**Fig 1. mNeonGreen-Tpm fusion proteins are functional and restore normal growth and actin cytoskeleton organization in Δtpm1 cells. (A)** Representative time-lapse montages of wildtype yeast cells expressing mNG-ASTpm1 (top) or mNG-ASTpm2 (bottom); scale bar - 3μm. **(B)** Representative images of cells of indicated yeast strains stained with Rhodamine-phalloidin; scale bar - 3μm. **(C)** Plot representing mean actin cable length per cell for

indicated strains as per experiment shown in **(B)**; n=15 cells per strain. **(D)** Representative images of indicated yeast strains expressing Om45-3xmCherry from the native locus; scale bar - 2µm. **(E)** Plot representing mean mitochondrial fragment count per cell for indicated strains as per experiment shown in S1G, n=16 cells per strain.(Box represents 25th and 75th percentile, line represents median, whiskers represent minimum and maximum values; One-Way Anova with Tukey's Multiple Comparisons test was used in **(C)** and **(D)**, * p < 0.05, ** p < 0.01, *** p < 0.001, **** p < 0.0001).

and unacetylated forms of the fission yeast Tpm, Cdc8, show a formin isoform-dependent spatial sorting to distinct actin cable networks [31,45]. Contrasting results from studies in mammalian cells support both formin-mediated [31,32] or relative concentration-dependent sorting mechanisms [30]. The presence of two Tpm isoforms maintained at distinct expression levels (Tpm1 expressed ~5–6 fold higher than Tpm2) [16,64] and previous observations showing that Δtpm1 exhibits distinct genetic interactions with either formin - Bnr1 and Bni1 indicates a possible crosstalk between Tpm and formin isoforms in *S. cerevisiae* [38]. Tpm1 also increased Bni1-mediated nucleation of actin filaments in an *in vitro* assay without having any effect on Bnr1-mediated nucleation in the same study [38]. However, whether these differences observed *in vitro* translate to measurable effects and possible formin-based spatial sorting of Tpm1 and Tpm2 *in vivo* remains unclear. To answer these questions, we investigated if Tpm1 and Tpm2 preferred actin cables nucleated by a specific formin isoform - Bnr1 or Bni1. Bnr1 and Bni1 localize to the bud neck and bud cortex respectively from G1 till onset of cytokinesis where they nucleate distinct sets of actin filaments [65–69]. Localization of both Tpm1 and Tpm2 on actin cables could be observed in both Δbnr1 and Δbni1 cells (Fig 2A and 2D), qualitatively suggesting that both Tpm1 and Tpm2 bind to actin cables nucleated by either formin - Bnr1 and Bni1 indiscriminately.

To quantitatively assess for any preference shown by Tpm1 and Tpm2 for formins Bnr1 and Bni1, we then performed an extensive analysis of actin cable organization visualized by phalloidin staining (S3A and S3D Fig) and Tpm localization visualized by mNG-ASTpm1/2 fusion proteins in wildtype, Δbnr1, and Δbni1 cells expressed under their native promoters from an extra integrated copy at the *leu2* locus (Fig 2A and 2D). We rationalized that if either Tpm isoform had a preference for decorating filaments made by only one of the formins, its localization would be severely affected in cells lacking that particular formin isoform. For our analysis, we measured average actin and Tpm-bound actin cable length and number in these strains as a population average instead of simultaneous co-staining in single cells (Figs 2A, 2D, S3A and S3D) because mNG-ASTpm signal drastically weakened after fixation making it difficult to score properly for Tpm localization. Our analysis revealed that actin cable numbers dropped significantly upon deletion of either formin Bnr1 and Bni1 consistent with the role for both formins in actin cable nucleation (S3C and S3F Fig), while actin cable lengths in Δbnr1 and Δbni1 were in general similar wildtype cells (S3B and S3E Fig) which could be explained by the recently described phenomenon of actin cable length scaling with cell size in *S. cerevisiae* [56,57]. Both Tpm1- and Tpm2-bound cable length did not show a consistent signifiant change between wildtype, Δbnr1, and Δbni1 cells (Fig 2B and 2E), agreeing with previous observations that actin cable length scales with cell size in *S. cerevisiae* [56,57]. In general, we observed that Tpm/actin cable lengths in wildtype, Δbnr1, and Δbni1 were variable and did not follow a very consistent pattern, likely due to a combination of factors which include natural heterogeneity and experimental stochasticity (e.g., variations in staining efficiency). However, the average number of Tpm1- and Tpm2-bound cables per cell were consistently reduced in Δbnr1 and Δbni1 cells as compared to *wildtype* (Fig 2C and 2F) but the reduction was comcomitant with the reduction in actin cable numbers observed in Δbnr1 and Δbni1 (S3C and S3F Fig) and no major dependence for Tpm1 and Tpm2 on either formin was supported. While our analysis suggests no preference between Tpm and formin isoforms, it remains possible that the localization of Tpm1 and Tpm2 may inherently depend on the formin isoforms, and deletion of the formin proteins may cause a change in the observed localization patterns in wildtype vs formin deletion strains. To address this, we assessed localization of mNG-ASTpm1 and mNG-ASTpm2 in wiildtype cells and clearly observed that mNG-ASTpm1 and mNG-ASTpm2 decorate actin cables in both the bud and mother compartments (S3G Fig). The ratio of Bud/Mother fluorescence in wildtype cells was very similar for both mNG-ASTpm1 and mNG-ASTpm2 suggesting that both Tpm1 and Tpm2 have very similar distributions among mother and bud

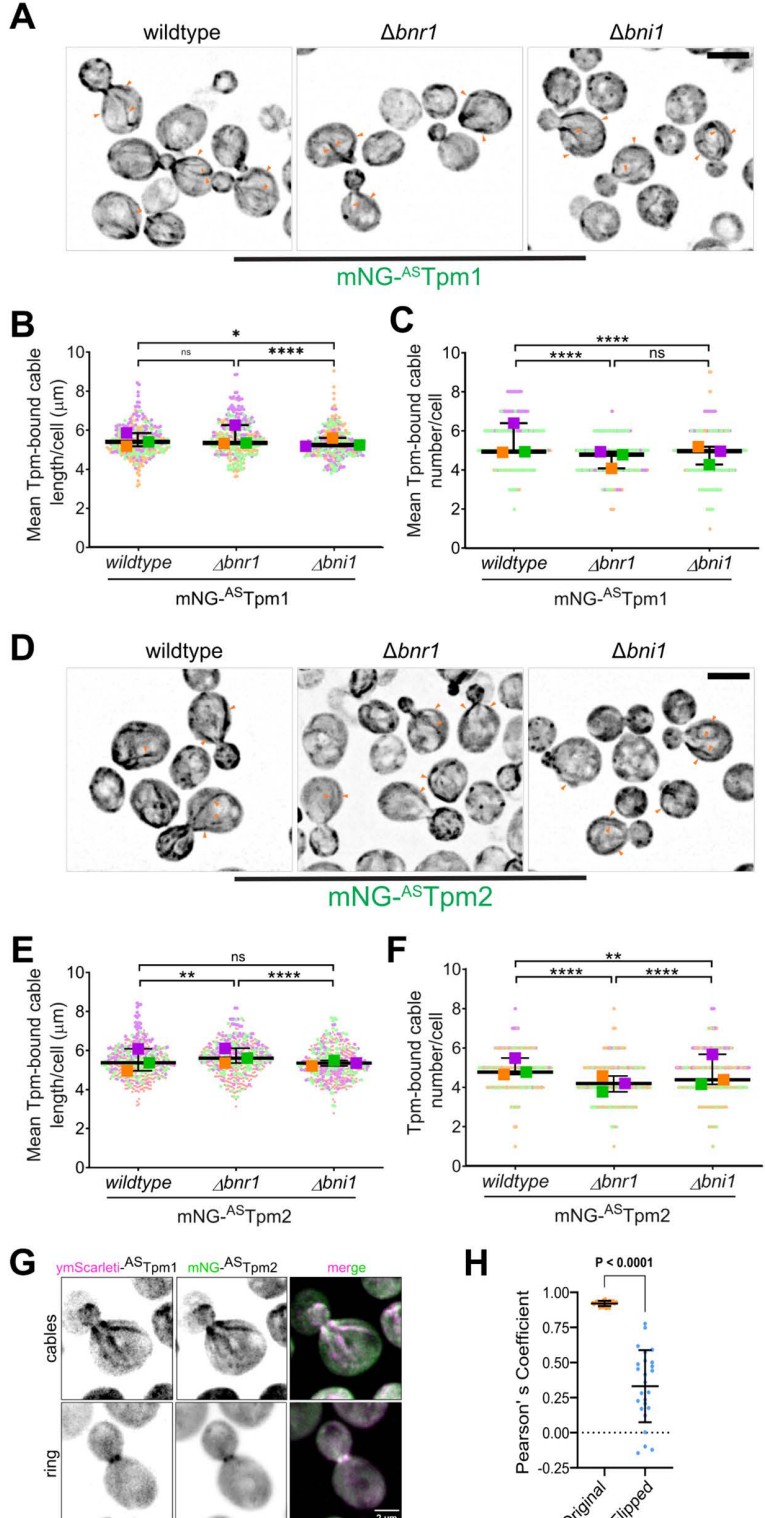

**Fig 2. Tpm1 and Tpm2 colocalize and indiscriminately bind to actin filaments nucleated by formin isoforms, Bnr1 and Bni1. (A)** Representative images of cells of wildtype, Δ*bnr1*, and Δ*bni1* cells expressing mNG-<sup>AS</sup>Tpm1 fusion protein; scale bar - 3μm. **(B)** Superplot representing mean

Tpm-bound cable length per cell in wildtype, Δ*bnr1*, and Δ*bni1* cells expressing mNG-<sup>AS</sup>Tpm1; n = 100 cells for each strain per replicate, N = 3. **(C)** Superplot representing mean Tpm-bound cable number per cell in wildtype, Δ*bnr1*, and Δ*bni1* cells expressing mNG-<sup>AS</sup>Tpm1; n = 100 cells for each strain per replicate, N = 3. **(D)** Representative images of cells of wildtype, Δ*bnr1*, and Δ*bni1* cells expressing mNG-<sup>AS</sup>Tpm2 fusion protein; scale bar - 3μm. **(E)** Superplot representing mean Tpm-bound cable length per cell in wildtype, Δ*bnr1*, and Δ*bni1* cells expressing mNG-<sup>AS</sup>Tpm2; n ≥ 100 cells for each strain per replicate, N = 3. **(F)** Superplot representing mean Tpm-bound cable number per cell in wildtype, Δ*bnr1*, and Δ*bni1* cells expressing mNG-<sup>AS</sup>Tpm2; n ≥ 100 cells for each strain per replicate, N = 3. **(G)** Representative images of wildtype yeast cells co-expressing ymScarletI-<sup>AS</sup>Tpm1 and mNG-<sup>AS</sup>Tpm2; scale bar - 3μm. **(H)** Plot depicting Pearson's Correlation Coefficient calculated for cells co-expressing ymScarletI-<sup>AS</sup>Tpm1 and mNG-<sup>AS</sup>Tpm2. Measurement was done on the original images with both channels in original orientation and with one of the channels flipped horizontally or vertically as a control; n = 25 cells. (Superplots represent datapoints and means from three independent biological replicates marked in different colours; * $p < 0.05$, ** $p < 0.01$; Kruskal-Wallis test with Dunn's Multiple Comparisons test was used in **(B)**, (C), (E), (F); * $p < 0.05$, ** $p < 0.01$, *** $p < 0.001$, **** $p < 0.0001$.

compartments in wildtype cells (S3H Fig). The ratio of Bud/Mother fluorescence increased in Δ*bnr1* cells and decreased in Δ*bni1* cells as compared to wildtype cells for both mNG-<sup>AS</sup>Tpm1 and mNG-<sup>AS</sup>Tpm2 (S3H Fig), consistent with the fact that Bnr1 nucleates cables only in the mother compartment and all cables present in the bud compartment are nucleated by Bni1 (S3A and S3D Fig). These observations combined with our live-cell imaging data (Fig 1A) suggests that Tpm1 and Tpm2 bind to cables nucleated by Bnr1 and Bni1 in wildtype cells and suggests against the possibility that they may show a preference for either formin only in wildtype cells. Thus, Tpm1 and Tpm2 do not show preference for binding to cables in either bud or mother compartment contrary to another actin cable-binding protein, Abp140, which preferably bind to actin cables only in the mother cell compartment [70].

Since, mNG-<sup>AS</sup>Tpm1 is functional when expressed under native Tpm1 promoter (S1D, S2C and S2D Figs), we also conducted our analysis in strains lacking endogenous Tpm1 and expressing only mNG-<sup>AS</sup>Tpm1 to eliminate any effects of increased levels of functional Tpm1 protein. We found that mNG-<sup>AS</sup>Tpm1 still localized to actin cables in wildtype, Δ*bnr1*, and Δ*bni1* cells (S4A Fig). Tpm-bound actin cable length and Tpm-bound actin cable number was decreased in Δ*bnr1* and Δ*bni1* cells as compared to wildtype (S4B and S4C Fig), which was again concomitant with the decrease in actin cable length and number (S4D, S4E and S4F Fig), thus, not supporting any Tpm-formin preference. Interestingly, Tpm-bound actin cable length seemed to be lower than actin cable length in this experiment suggesting that mNG-Tpm fusion proteins may show better decoration and signal on actin cables when expressed in the presence of the endogenous Tpm protein. This could be caused due to the possibility of heterodimer formation between tagged and untagged Tpm monomers and result in higher binding affinity as to that of a homodimer of two tagged Tpm monomers. Overall, this analysis supports our conclusion that Tpm1 binds to filaments made by either formin isoform. A similar analysis for mNG-<sup>AS</sup>Tpm2 was not possible due to it not being functional when expressed from its native promoter at levels similar to Tpm2 endogenous levels (S2C and S2D Fig).

Thus, our quantitative analysis overall suggests that Tpm1 and Tpm2 binding to actin cables is unaffected by identity of the formin isoforms- Bnr1 and Bni1, implicating that they indiscriminately bind to both Bnr1- and Bni1-nucleated actin cables in *S. cerevisiae.*

Lastly, we assessed if Tpm1 and Tpm2 localize to the same set of actin cables *in vivo* by co-expressing ymScarletI-<sup>AS</sup>Tpm1 and mNG-<sup>AS</sup>Tpm2 fusion proteins. We observed that both ymScarletI-<sup>AS</sup>Tpm1 and mNG-<sup>AS</sup>Tpm2 localized to the same set of actin cables (Fig 2G) in cells which was also coroborrated by the high degree of Pearson's correlation coeffcient (Fig 2H). These results indicate that they may either co-polymerize on a single actin filament or are bound to different actin filaments bundled together into an actin cable.

Overall, our qualitative and quantitative analysis suggests that Tpm1 and Tpm2 indiscriminately bind to actin filaments nucleated by either formin, Bnr1 and Bni1, and do not exhibit spatial sorting to distinct actin cable networks in *S. cerevisiae*. Since, Tpm1 and Tpm2 have differential expression levels in cells, their relative local concentrations may be the major factor in determining their localization to actin filament networks in *S. cerevisiae* and governing their functions.

PLOS Genetics

## Tpm1 and Tpm2 localize to the fusion focus in a Bnr1-independent manner in mating yeast cells

Polarized actin assembly is required for growth of the pheromone-induced mating projection (shmoo) in yeast cells [71–77]. Similar to polarized bud growth, formin-made actin cables act as tracks for vesicles to be delivered to the tip of the mating projection where coordinated exo- and endo- cytosis ensure polarized growth, membrane turnover, and cell wall remodelling [78–82]. It has been shown that Δtpm1 cells require much higher pheromone concentrations for shmoo formation as compared to wildtype cells [83]. We wondered if Tpm1 and Tpm2 both localized to the formin-made actin filaments at the mating projection and whether their localization was dependent on any particular formin isoform. To answer these questions, we mated wildtype *MAT-a* haploid yeast cells expressing either mNG-^ASTpm1 or mNG-^ASTpm2 with wildtype *MAT-α* yeast cells. Both mNG-^ASTpm1 and mNG-^ASTpm2 localized to the mating projection tip (Fig 3A and S3 and S4 Movies) and their level started gradually decreasing ~2–4 minutes before fusion of cells could be observed (Fig 3C). These results demonstrate that Tpm1 and Tpm2 localize to the mating fusion focus and suggest a role for them in stabilizing formin-based actin filaments during yeast mating as also previously observed for *S. pombe* Tpm ortholog Cdc8 [44]. Previous studies with *S. cerevisiae* have shown that the pheromone induced shmoo formation is dependent only on the formin Bni1 which is recruited via Bil2 and requires Bud6 activity [72,84,85]. So, we performed an experiment where we mated Δbnr1 cells expressing either mNG-^ASTpm1 or mNG-^ASTpm2 with wildtype cells of opposite mating type. We observed that both mNG-^ASTpm1 and mNG-^ASTpm2 localized to the mating projection tip with similar kinetics in Δbnr1 cells similar to wildtype cells (Fig 3B and 3D), suggesting that both Tpm1 and Tpm2 decorate Bni1-made actin filaments at the mating projection during the mating pheromone response in *S. cerevisiae.* These observations suggest a role for Tpm1 and Tpm2 in stabilization of Bni1-made actin cables during budding yeast mating and further strengthen our conclusion that Tpm1 and Tpm2 indiscriminately bind to formin-made filaments irrespective of the identity of the formin isoform in *S. cerevisiae*.

## Tpm1 and Tpm2 protect equally well from cofilin-mediated actin filament severing

Tpm1 and Tpm2 isoforms are believed to play both shared and distinct roles in *S. cerevisiae* [16,39]. The major isoform Tpm is expressed 5–6 fold higher than Tpm2 and its absence leads to a near complete loss of actin cables [15,38,83]. A previous study reported that overexpression of Tpm2 could not restore actin cables in a Δtpm1 strain suggesting that Tpm2 and Tpm1 may have distinct functions [16]. We were intrigued by this observation as Tpm1 and Tpm2 have identical subcellular localization and dynamics as revealed by our experiments, which suggests a strong possibility for overlap of function (Figs 1A, 2G, 2H and S3G). Thus, we decided to revisit Tpm2 functionality and determine its subcellular roles in maintaining the actin cytoskeleton. Firstly, to test whether Tpm1 and Tpm2 impart similar protection against cofilin-mediated severing, we performed an *in vitro* TIRF imaging [86,87] with purified yeast tropomyosins (Tpm1, Tpm2), human Tpm1.7 [88] (positive control) and budding yeast Cofilin (Cof1) (Fig 4A). We found that Tpm1.7, Tpm1 and Tpm2 coated-actin filaments displayed significantly lower severing events per unit time and length (Fig 4B) and resulted in higher filament lengths after 2 min of cofilin addition as compared to uncoated control filaments (Fig 4C). However, Tpm1 and Tpm2 showed significantly less protection than the human Tpm1.7, suggesting that yeast Tpm allow for faster turnover of actin filaments via cofilin-mediated severing. These observations suggest that Tpm1 and Tpm2 display similar biochemical activities of actin filament protection from severing action of cofilin (Fig 4D), and hint towards an potential protective role of Tpm2 in cells.

## Tpm2 overexpression can compensate for loss of Tpm1 function *in vivo*

To coroborate our in-vitro findings and test whether Tpm2 could peform similar functions as Tpm1 in vivo, we assessed Tpm2's ability to rescue defects observed in cells lacking Tpm1. First, we checked whether Tpm2 could complement growth defects caused by loss of Tpm1, we overexpressed Tpm2 in a Δtpm1 strain using a high-copy number plasmid (pRS425$_{(H)}$) and surprisingly, found that Tpm2 overexpression could restore growth rates similar to wildtype levels in

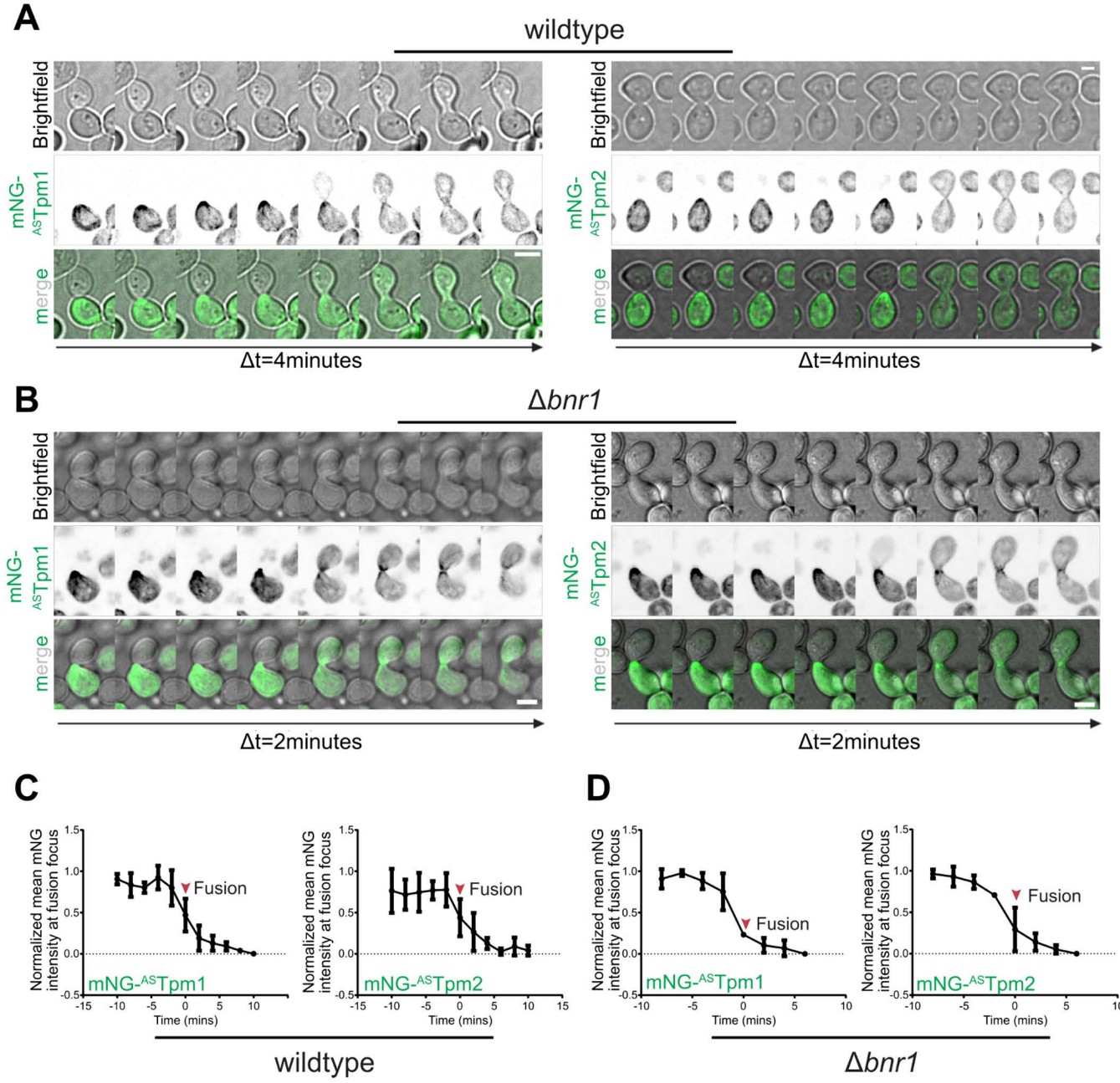

**Fig 3. Tpm1 and Tpm2 co-localize at the fusion focus in mating yeast cells in a Bni1-dependent manner. (A)** Representative time-lapse montages of haploid *MATa* type wildtype yeast cell expressing mNG-^ASTpm1 (left panel) or mNG-^ASTpm2 (right panel) mating with haploid *MATα* wildtype yeast cell; scale bar – 4μm. **(B)** Representative time-lapse montages of haploid *MATa* type Δ*bnr1* yeast cell expressing mNG-^ASTpm1 (left panel) or mNG-^ASTpm2 (right panel) mating with haploid *MATα* wildtype yeast cell; scale bar – 4μm. **(C-D)** Line plot showing protein accumulation kinetics of mNG-^ASTpm1 or mNG-^ASTpm2 at the mating projection tip in wildtype **(C)** or Δ*bnr1* **(D)** cells. t = 0 represents cell fusion (n = 3 fusion events/strain).

Δ*tpm1* cells (S5A Fig). This intriguing finding, contrary to previous results reported by Drees et al., (1995) [16], led us to investigate whether Tpm2 overexpression could also restore actin cables in Δ*tpm1* cells. Interestingly, we observed that Tpm2 overexpression could rescue actin cables similar in length and number to wildtype in ~80% of Δ*tpm1* cells (S5B,

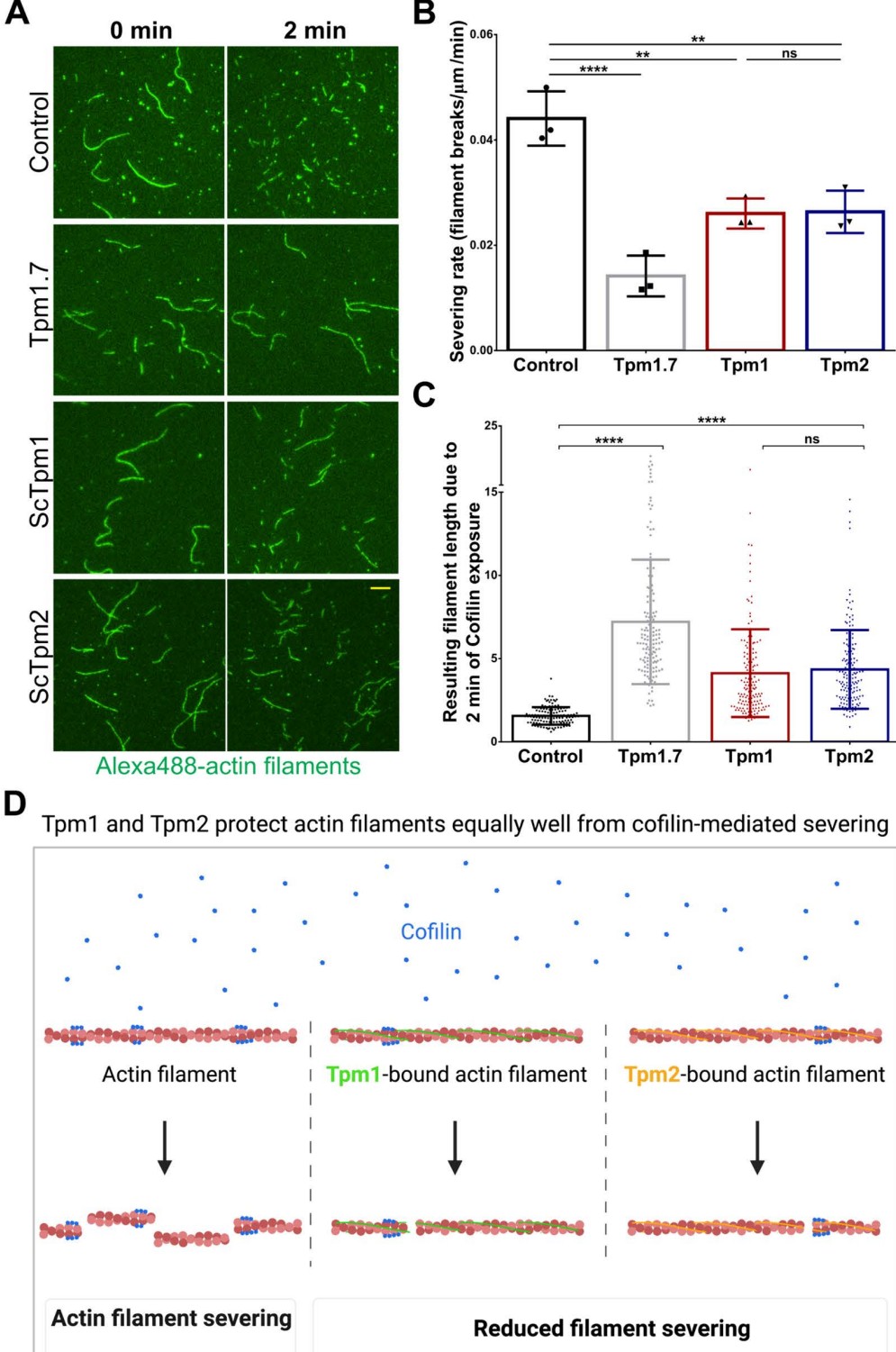

**Fig 4. Tpm2 can protect actin filaments from cofilin-mediated severing *in vitro*. (A)** Representative time-lapse images of the actin filaments before (left) and 2 min after the addition of recombinant ScCofilin (right) in absence or presence of different Tpm isoforms; scale bar - 10µm. **(B)** Plot

representing number of severing events per μm of actin filaments per minute. Data shown for independent trials; n ≥ 150 filaments in each condition, N = 3. **(C)** Plot representing actin filament lengths after 2 minutes of addition of Cof1 to the samples in absence (control) or presence of specified tropomyosin; n = ~200 filaments, N = 3. **(D)** Schematic model showing that Tpm1 and Tpm2 protect actin filaments equally well from cofilin-mediated actin severing in vitro, created using BioRender, https://BioRender.com/792vjsp. (Box represents 25th and 75th percentile, line represents median, whiskers represent minimum and maximum value; One-Way Anova with Tukey's Multiple Comparisons test was used in **(B)** and **(C)** * p < 0.05, ** p < 0.01, *** p < 0.001, **** p < 0.0001).

S5C, S5D and S5E Fig). We confirmed Tpm2 overexpression using RT-qPCR data which revealed a ~19 fold higher Tpm2 mRNA abundance after expression through high-copy number plasmid ($pRS425_{(H)}$-$^{pTpm2}Tpm2$) as compared to Tpm2 mRNA levels in wildtype and Δtpm1 cells (S5F Fig). This data clearly suggests that in contrary to previous understanding [16], elevated levels of Tpm2 can organize polarized actin cables in the absence of Tpm1, potentially by protecting actin cables from the severing action of cofilin. Our findings differ from a previous study, which reported that Tpm2 overexpression using a galactose-inducible high-copy plasmid could not rescue the phenotypes of Δtpm1 cells [16]. To verify our results and assess whether this discrepancy was due to excessive overexpression from the GAL1 promoter which could potentially be toxic for cells, we expressed either Tpm1 or Tpm2 from a GAL1-driven high-copy plasmid in our strain. We found that GAL1-driven overexpression of both Tpm1 and Tpm2 completely rescued the growth and actin cable defects in Δtpm1 cells (S6A and S6B Fig). These results indicate that strong overexpression from the GAL1 system is not responsible for the difference in observations and further support our conclusion that Tpm1 and Tpm2 are fully functionally redundant for actin cable stability. This represents a previously unrecognized activity for Tpm2 and raises further interesting questions about the shared and distinct roles of Tpm isoforms in *S. cerevisiae*.

## A ~ 2.5 fold increase in Tpm2 expression can restore actin cable organization in Δtpm1 cells

While overexpression of Tpm2 completely rescues defects observed in Δtpm1 cells, we next asked whether a moderate increase in Tpm2 protein levels could also have the same effect. To test this, we used low-copy plasmid expression of Tpm2 under native Tpm2 and Tpm1 promoters ($^{pTpm2}Tpm2$ and $^{pTpm2}Tpm1$) in Δtpm1 cells. We tested their ability to rescue growth of Δtpm1 cells and found that both $^{pTpm2}Tpm2$ and $^{pTpm1}Tpm2$ constructs restored wildtype growth levels in Δtpm1 cells (S7A Fig). Expression of $^{ptpm2}Tpm1$ and $^{pTpm1}Tpm1$ (positive control) also restored normal growth levels in Δtpm1 cells (S7A Fig). F-actin staining with Alexa488-phalloidin showed rescue of actin cables by all Tpm2 and Tpm1 expression constructs in ~90% Δtpm1 cells (Fig 5A and 5B) to similar length and number as the wildtype (Fig 5C and S7B). To determine the amount of Tpm2 mRNA after expression through low-copy plasmids, we performed RT-qPCR and found that Tpm2 mRNA levels showed ~3 fold average increase when expressed under the pTpm1 promoter and ~2.5 fold increase when expressed under its native *ptpm2* promoter, as compared to wildtype cells (S7C Fig). Wildtype and Δtpm1 cells had similar Tpm2 mRNA levels suggesting Tpm2 expression is not sensitive to loss of Tpm1 (S7C Fig). In the absence of specific antibodies against Tpm1 and Tpm2, we used the functionality of mNG-$^{AS}$Tpm fusion proteins to rescue phenotype in Δtpm1 cells and ascertain that increased protein levels of Tpm2 are reponsible for the rescue. High-copy plasmid ($pRS425_{(H)}$) expression of both mNG-$^{AS}$Tpm1 and mNG-$^{AS}$Tpm2 under their native promoters rescued the severe growth defects of Δtpm1 cells at 37°C, but only mNG-$^{AS}$Tpm1 and not mNG-$^{AS}$Tpm2 rescued the defect when expressed from an integrated plasmid copy ($pRS305_{(I)}$) under their native promoters (S8A Fig). Quantification of mean mNG signal intensity per cell revealed that the highest expression was observed in $pRS425_{(H)}$-$^{pTpm1}mNG$-$^{AS}Tpm1$ followed by $pRS425_{(H)}$-$^{pTpm2}mNG$-$^{AS}Tpm2$, $pRS305_{(I)}$-$^{pTpm1}mNG$-$^{AS}Tpm1$ and $pRS305_{(I)}$-$^{pTpm2}mNG$-$^{AS}Tpm2$ (S8B and S8C Fig), consistent with our earlier observation that mNG-Tpm2 fusion is functional at higher expression levels as compared to native untagged Tpm2 (S2C and S2D Fig). We also performed western blotting with anti-mNG antibody to detect protein levels (S8D Fig) and found that $pRS425_{(H)}$-$^{pTpm1}mNG$-$^{AS}Tpm1$ had the highest expression levels and $pRS425_{(H)}$-$^{pTpm2}mNG$-$^{AS}Tpm2$

PLOS Genetics

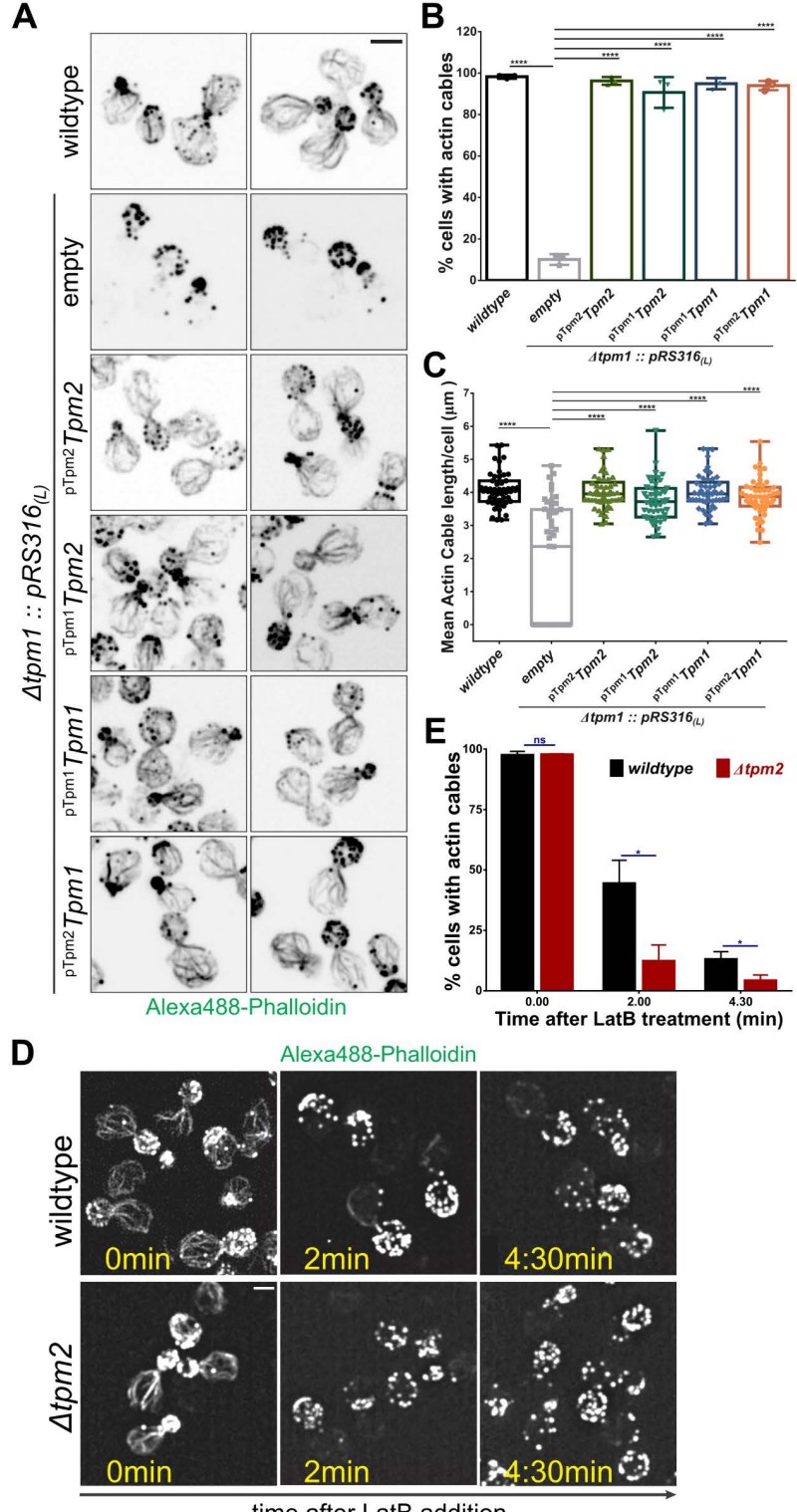

Alexa488-Phalloidin

**Fig 5. Increased Tpm2 expression can restore actin cables in Δ*tpm1* cells. (A)** Representative images of indicated yeast strains stained with Alexa488-phalloidin; scale bar – 3μm. **(B)** Plot representing mean percentage of cells with detectable actin cables in indicated yeast strains averaged over 3 biological replicates; n > 150 cells for each strain per replicate, N = 3. **(C)** Plot representing mean actin cable length per cell in the indicated yeast

strains as per experiment in **(A)**; n = 50 cells for each strain. **(D)** Representative time-course images of wildtype and Δ*tpm2* cells treated with Latrunculin B (67μM) and stained with Alexa488-phalloidin; scale bar - 2μm. **(E)** Plot representing mean percentage of cells with detectable actin cables in indicated yeast strains averaged over 3 biological replicates; n > 125 cells per strain per replicate, N = 3. (Box represents 25th and 75th percentile, line represents median, whiskers represent minimum and maximum value; One-Way Anova with Tukey's Multiple Comparisons test was used in **(C)** and Unpaired two-tailed t-test was used in **(E)**, * $p < 0.05$, ** $p < 0.01$, *** $p < 0.001$, **** $p < 0.0001$).

and pRS305$_{(l)}$-$^{pTpm1}$mNG-$^{AS}$Tpm1 only expressed to ~20% and ~10% of that level (S8E Fig)**.** Protein levels were not detectable in pRS305$_{(l)}$-$^{pTpm2}$mNG-$^{AS}$Tpm2 containing Δ*tpm1* cells (S8D Fig) and in conjunction with our mean intensity analysis (S8B and S8C Fig) explain the lack of rescue observed in Δ*tpm1* cells containing pRS305$_{(l)}$-$^{pTpm2}$mNG-$^{AS}$Tpm2. The inability to detect mNG-Tpm2 protein levels (S8D Fig) from low-copy plasmid expression may be due to the weak strength of the native Tpm2 promoter (pTpm2). Taken together, these data clearly suggest that it is the increase in mNG-$^{AS}$Tpm2 protein levels that lead to rescue of observed phenotype in Δ*tpm1* cells.

These observations suggest that Tpm2 contributes to actin cable stability in vivo after increased expression, which prompted us to test whether it provides any measurable protection to actin cables at its native expression levels. To test this, we studied actin cable sensitivity in the presence of actin monomer sequestering drug, Latrunculin B (LatB) in wildtype vs Δ*tpm2* cells. (Fig 5D). We found that Δ*tpm2* cells are significantly more sensitive to the depolymerizing effect of LatB than wildtype cells (Fig 5D and 5E), suggesting that Tpm2 does play previously unrecognized detectable roles in actin cable stability in conditions of stress and contributes to robustness of the actin cable network, even at its endogenous levels.

Together, these results show that even minor increases in Tpm2 expression can restore normal actin cable organization in Δ*tpm1* cells suggesting that Tpm2 protein is present at limiting levels but retains the function of actin cable protection from severing factors like cofilin *in vivo* and *in vitro*. Tpm2 can, therefore, act as a safeguard mechanism for cells to protect against defects in Tpm1, providing functional redundancy and robustness against cellular stressors for actin cable maintenance in *S. cerevisiae.*

### Tpm2 can solely organize a functional actin cable network in *S. cerevisiae*

Actin cables in *S. cerevisiae* are required for functions such as anterograde transport of vesicles and organelles [22,38,41,65,66,83], maintenance of mitochondrial inheritance and morphology [53,55,89,90], and clearance of protein aggregates from the bud [91,92] while retaining damaged mitochondria in the mother via retrograde actin cable flow (RACF) [40,93]. We asked whether Tpm2 organized actin cables could perform all these functions and if Tpm1-organized and Tpm2-organized actin cable networks had functional differences. We systematically analyzed these functional readouts using fluorescent reporters in wildtype, Δ*tpm1*, and Δ*tpm1* cells expressing Tpm2 in low-copy number and high-copy number plasmids. We used exogenously expressed ymScarleti-Sec4 [38] integrated into the *ura3*/*leu2* locus to assess vesicle targeting to the bud via Myo2-dependent transport on actin cables [22] (Figs 6A and S9A). The ratio of bud-to-mother Sec4 fluorescence was calculated for all strains and the results showed that vesicle targeting to the bud was restored to levels similar to wildtype cells by expression of both $^{pTpm2}$Tpm2 and $^{pTpm1}$Tpm2 constructs (Figs 6C and S9C). Tpm1 expression also rescued vesicle targeting as a positive control under pTpm1 and pTpm2 promoters (Figs 6A, 6C and S9C). These results suggest that both Tpm2 and Tpm1 can individually facilitate normal anterograde flow of cargo and vesicles to the growing bud via facilitation of Myo2 processivity on actin cables, in agreement with previous *in vitro* experiments where both Tpm1 and Tpm2 increased processivity of the type-V myosin, Myo2, on coated actin filaments [41].

Next, we assessed whether the hyper-fragmented mitochondrial morphology, a characteristic of Δ*tpm1* cells [53,55], could be restored by exogenous expression of Tpm2. For visualization of mitochondrial morphology, we used Su9-mNeonGreen [94] (subunit 9 of the F1-F0 ATPase fused to mNeonGreen) construct integrated at the *leu2*/*ura3* locus

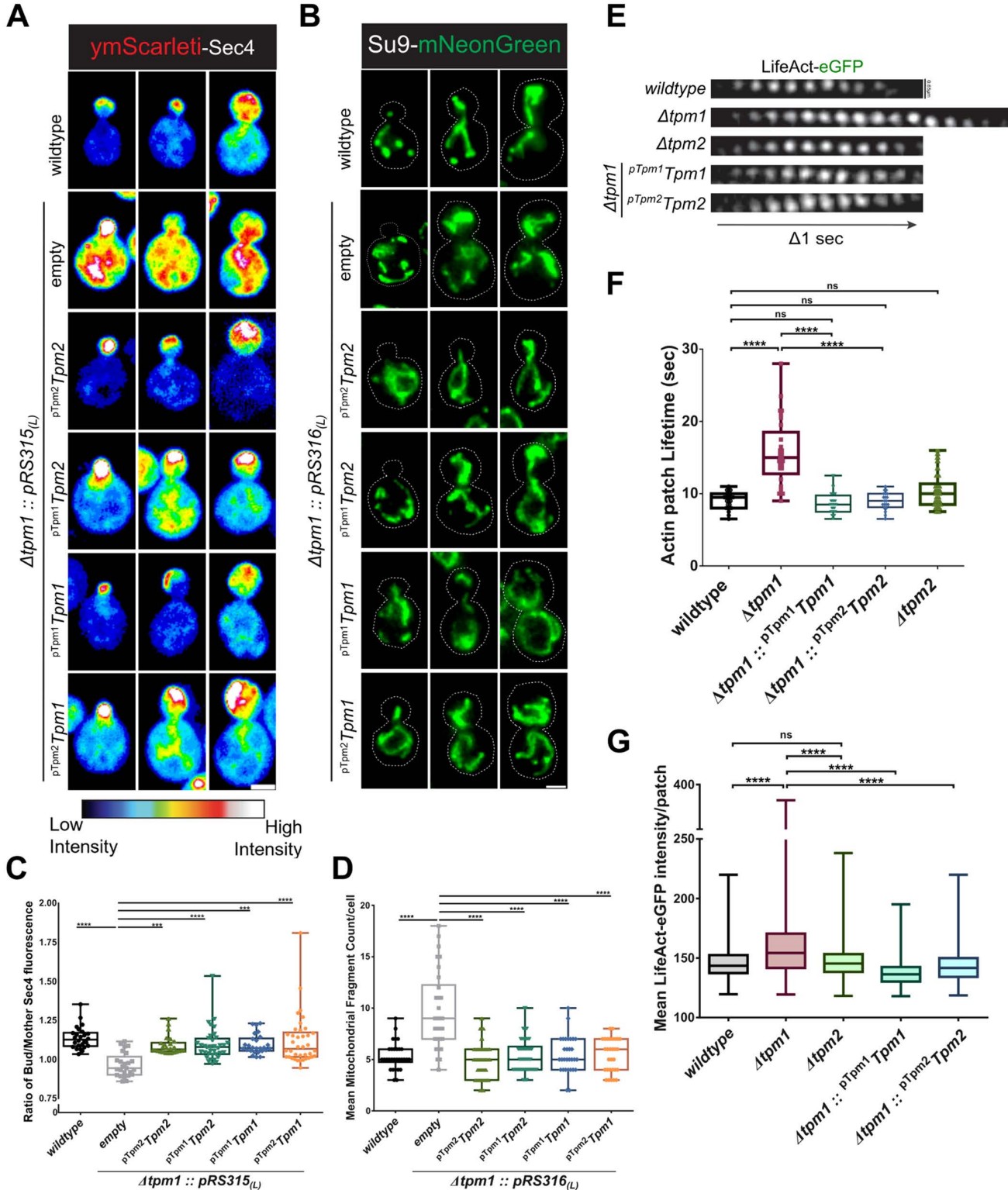

**Fig 6. Tpm2 can independently organize a functional actin cable cytoskeleton in Δtpm1 cells. (A)** Representative images of indicated yeast strains expressing ymScarleti-Sec4; scale bar – 3μm. **(B)** Representative images of indicated yeast strains expressing Su9-mNeonGreen; Scale bar – 3μm. **(C)** Plot representing ratio of bud/mother ymScarleti-Sec4 fluorescence per cell in the indicated yeast strains; n ≥ 25 cells per strain. **(D)** Plot

representing mean mitochondrial fragment count per cell in the indicated yeast strains; n ≥ 30 cells per strain. **(E)** Representative time-lapse images of actin patch dynamics in cells with indicated genotypes expressing LifeAct-eGFP, scale bar – 0.65µm. **(F)** Box and Whiskers plot showing actin patch lifetimes in the indicates yeast strains, (n > 20 patches/strain). **(G)** Box and Whiskers plot showing mean LifeAct-eGFP fluorescence in the indicates yeast strains, (n ≥ 735 cells/strain). (Box represents 25th and 75th percentile, line represents median, whiskers represent minimum and maximum value; One-Way Anova with Tukey's Multiple Comparisons test was used in **(C)**, Kruskal-Wallis test with Dunn's multiple comparisons test was used in **(D)**, **(G)**, and **(F)**; * p < 0.05, ** p < 0.01 * p < 0.05, ** p < 0.01, *** p < 0.001, **** p < 0.0001).

and quantified average mitochondrial fragment number per cell (Figs 6B and S9B). Δ*tpm1* cells showed increased mean mitochondrial fragment number as compared to wildtype while expression of Tpm2 or Tpm1 under native promoters in low copy or high copy plasmids restored the average mitochondrial fragment number to wildtype levels (Figs 6D and S9D). These results suggest that Tpm2-bound actin cables can interact and crosstalk normally with mitochondria to maintain its normal morphology. The underlying mechanisms of how actin cables control mitochondrial morphology in budding yeast remain largely unknown and future studies are required to understand this enigmatic phenomenon.

Tpm2 has been previously implicated in negative regulation of retrograde actin cable flow (RACF) rates, possibly via its inhibitory effect on binding of the type-II myosin (Myo1) to actin filaments [39,40]. We independently replicated this experiment using LifeAct-eGFP [70,95] as an actin cable marker and corroborated that RACF rates are increased in the absence of Tpm2 and exogenous expression of Tpm2 and not Tpm1 in Δ*tpm2* cells restores RACF rates to wildtype levels (S9E and S9F Fig). RACF rates were similar to wildtype in Δ*tpm1* cells in which Tpm2 was expressed to restore actin cable suggesting again that loss of Tpm1 did not affect RACF rates in cells (S9F Fig). These observations suggest that Tpm2 has gained distinct regulatory function from Tpm1 in negatively regulating RACF while retaining its actin protective activity.

## Global Tropomyosin levels are important for proper function and homeostasis between linear and branched actin in *S. cerevisiae*

Homeostasis between linear and actin networks in cells is maintained by various actin-binding proteins that coordinate and compete to generate actin filament network diversity in cells [96–101]. In yeast, formin proteins nucleate and maintain linear actin filaments which bundle and form cables [65,66] while Arp2/3 complex generates branched actin filaments which form a dense actin network at sites of endocytosis called actin patches [102,103]. Actin cables and patches interplay with each other while competing for the available cytoplasmic pool of actin monomers [97,99,104,105]. Tpm1 and Tpm2 localize to actin cables (Fig 1A) and are majorly excluded from actin patches due to the activity of Fimbrin (Sac6) which majorly localizes to actin patches [52,96]. We have previously shown, using mNG-Tpm fusion proteins, that Tpm1 and Tpm2 localize to actin patches in cells lacking Fimbrin [44] as previously also observed in *S. pombe* [52,96] and recent studies observed similar targeting of Tpm and formin isoforms to actin patches in the absence of Capping protein suggesting a tight regulation of protein composition between these two networks in yeast [98,106]. Tpm loss or overexpression has been shown to opposingly affect total number of actin patches [99]. To better understand how Tpm isoforms affect actin patch dynamics during endocytosis, we imaged actin assembly and disassembly at actin patches using LifeAct-eGFP as a marker. We observed that loss of actin cables observed in Δ*tpm1* cells could lead to effects on actin patch-mediated endocytosis and found that actin patch lifetime was significantly increased in Δ*tpm1* cells (~15 sec) as compared to wildtype and Δ*tpm2* cells (~10 sec) (Fig 6E and 6F). The increased patch lifetime was restored to wildtype levels upon exogenous expression of either Tpm1 or Tpm2 from a low-copy plasmid (Fig 6E and 6F), suggesting that the observed increase is caused due to the global decrease in Tpm levels and loss of actin cables, and not specifically due to loss of Tpm1. We next tested whether the loss of actin cables observed in Δ*tpm1* cells lead to increased accumulation of actin at the actin patches and found that mean LifeAct-eGFP intensity at actin patches was higher in Δ*tpm1* cells as compared to wildtype and Δ*tpm2* cells (Fig 6G). The mean intensity was restored to wildtype levels upon exogenous

expression of Tpm1 and Tpm2 in Δ*tpm1* cells (Fig 6G), suggesting that reduction in global Tpm levels leads to dysregulation of actin cables, increased actin accumulation at the actin patches, eventually leading to abnormal endocytosis dynamics. These results highlight the isoform independent importance of Tpm in maintaining normal balance and homeostasis between actin networks in a common cytoplasmic volume in *S. cerevisiae* and opens avenues for future study of actin network crosstalk via effects of actin-binding protein in yeast and other eukaryotes.

Overall, our results suggest that Tpm2 can independently organize a functional actin cytoskeleton in *S. cerevisiae*. Our work highlights a previously unappreciated function of Tpm2 and demonstrates functional redundancy between Tpm isoforms despite evolutionary divergence and evolution of distinct functions such as RACF regulation. This functional redundancy may benefit cells under stress conditions and contribute to robust cell survival and viability.

## Discussion

### Making functional tropomyosin fusion proteins

In this study, we characterize fluorescently tagged-tropomyosin fusion proteins for functionality, which make it possible to probe tropomyosin isoform localization, functions, and competition with other actin-binding proteins in live cells. Building on our previous work [44], we show that mNG-Tpm1 and mNG-Tpm2 are functional and can restore viability of Δ*tpm1*Δ*tpm2* cells which display synthetic lethality (S2C Fig). To our knowledge, this represents the first demonstration and characterization of tagged tropomyosin proteins which near completely restore growth and actin organization in cells lacking native Tpms (Figs 1, S1 and S2). The good functionality and ability to clearly report Tpm localization makes mNG-Tpm fusion proteins a valuable resource for future research on Tpm isoforms across fungal and mammalian species. Previous work in *S. pombe* had also identified N-terminus of *S. pombe* Tpm (Cdc8) as being favorable for tagging while the C-terminal fusion rendered the protein completely non-functional [46]. While the N-terminally tagged Cdc8 could partially restore growth in a *cdc8* temperature-sensitive (ts) mutant bearing *S. pombe* strain, its localization was only clearly seen only on the contractile actomyosin ring and not on formin-made actin cables, again suggesting loss of function due to the tagging. Studies with the filamentous fungi *Aspergillus nidulans* have also used a variety of tagged-Tpm constructs but the used constructs could not sustain viability on their own [107,108]. A key difference between our study and the previous studies is the presence of a 40-amino acid linker between the fluorescent tag and the Tpm protein [44], which may allow for proper functionality, perhaps via proper head-to-tail contacts between Tpm dimers. This is evident by the clear localization observed on actin cables with our mNG-Tpm fusion proteins and rescue of growth and actin cable cytoskeleton observed in our study (Figs 1A-D, S1 and S2), both of which is not possible in case of direct N-terminal tagging of Tpm [46,109]. N-terminal tagging also has its caveats as it may block N-terminal acetylation of Tpm which is essential for normal Tpm binding to F-actin in cells [47,48] and also for regulation of other actin-binding proteins such as myosin-II and myosin-V [24,45]. To overcome this limitation, we generated mNG-[AS]Tpm fusion proteins which contain an Alanine-Serine (-AS-) dipeptide which is commonly used to mimic the effect of N-terminal acetylation. The use of -AS- dipeptide is routinely used to purify Tpm proteins from *E. coli* as it restores normal binding affinity [50] and its addition to the N-terminus of Tpm in the mNG-Tpm fusion improved their functionality by restoring normal length of actin cables in our experiments (Fig 1B and 1C). The observed effect may be due to increased binding affinity of -AS- containing Tpm for F-actin in cells as compared to unacetylated tagged Tpm and suggests a role for Tpm activity in cell-size dependent scaling of actin cables in yeast. Interestingly, the fact that partial length actin cables can near completely restore growth and mitochondrial morphology suggests that these functions may majorly depend on the cables being present, polarized and properly bundled, instead of them being full length. Previous observations that vesicle transport to the growing daughter cell is hampered in mutants where actin cable architecture is abnormal supports this view [110,111]. Previous work suggested that acetylation decreases the destabilizing effect of the positive charge of the N-terminus on the coiled-coil 'a' position where a hydrophobic residue normally resides [50]. A later study proposed that the -AS- dipeptide ensures that the Met1 stays in

an alpha-helical structure by introducing stabilizing hydrogen bonds between the Ser and Met residues [112]. Thus, -AS-dipeptide addition likely does not mimic N-terminal acetylation structurally but acts via maintaining N-terminus stability. Future work is required to understand in detail the effects of the linker and -AS- dipeptide on Tpm biochemical activity and effect on interactions with other actin-binding proteins. In our experiments, cells expressing mNG-ASTpm1 in addition to the endogenous Tpm vs cells only expressing mNG-ASTpm1 as a sole copy, we observed that the presence of endogenous Tpm allows for better decoration of actin cables (S4 Fig). This can be explained by the fact that mNG-Tpm fusion protein monomers can make a heterodimer with the native Tpm monomer or make a homodimer with another mNG-Tpm monomer, and the heterodimer may have better binding affinity to actin cables as compared to the homodimer due to lower steric hindrance. Also, since we did not see any abnormal effect due on cell growth or the actin cytoskeleton due to the additional expression of mNG-Tpm proteins in our study, it might be beneficial in most contexts to use these fusion proteins as additional copies in the presence of the endogenous Tpm.

## Spatial sorting of Tpm isoforms

The molecular basis of Tpm isoform sorting has remained an unresolved enigma in the field. Various studies over the years have implicated different mechanisms for spatial sorting of Tpm isoforms across different model systems. While *S. pombe* Tpm (Cdc8) exhibits sorting based on preference between N-terminal acetylation status and formin isoforms [45], mammalian Tpm isoforms have been proposed to sort through either formin-based nucleation [32] or relative concentration dependent mechanisms [30]. However, N-terminal acetylation did not bias localization of Cdc8 to filaments made by a particular *S. pombe* formin isoform in a minimal in-vitro reconstituted set-up [113], in contrast to the earlier *in vivo* observations [45], suggesting that interaction of the distinct Cdc8 forms (unacetylated vs acetylated) with distinct formins may not be enough for the spatial sorting observed and may require additional cellular factors. In this study, we utilized our functional mNG-Tpm constructs to address this long-standing question of Tpm isoform sorting in the model *S. cerevisiae* and find a formin-independent mode of binding of Tpm isoforms to actin cables (Figs 2, S3, S3 and 7A), in contrast to the formin-dependent modes observed in *S. pombe* [31]. We also performed first-ever dual-color imaging of Tpm1 and Tpm2 simultaneously in live cells which highlights the multiplexing possibilities of our tagging strategy to study Tpm isoform diversity (Fig 2G and 2H). Strikingly, while *tpm1* shows distinct genetic interactions with formins Bnr1 and Bni1 [38], it did not show any preference for filaments made by any one of these isoforms in our study. This seemingly contrasting results could be explained by the observations from *Shin et. al.,* 2018 [114], where difference in activities of Bnr1 and Bni1 in the presence of limiting levels of Tpm2 inside cells could explain why loss of Bnr1 or Bni1 in Δ*tpm1* cells has opposite effects on cell growth and fitness. Development of probes enabling measurement of formin activity in cells could help confirm this hypothesis in the future. While it is possible that N-terminal tagging may influence formin-based Tpm isoform sorting, C-terminally tagged constructs cannot currently be used to address this question due to their non-functional nature. Thus, there is still scope for developments and improved strategies for Tpm visualization to revisit these interesting questions. Overall, our data suggests distinct evolutionary adaptations and strategies for Tpm isoform sorting across species, which may be contingent on species-specific needs and adaptations.

## Shared and sistinct functions of Tpm1 and Tpm2

Our work also addresses the question of functional overlap and divergence of Tpm isoforms where we report a previously unknown role of the "minor" isoform Tpm2 in actin cable protection suggesting that Tpm2 has retained its actin protective function despite evolutionary divergence and gain of distinct regulatory activity in controlling retrograde actin cable flow as compared to Tpm1 [39] (Figs 4, 5, 6, S9E-F and 7B). We find that Tpm1 and Tpm2 protect equally well from cofilin *in vitro* (Fig 4A-D), suggesting that their binding sites on F-actin may overlap with cofilin-binding site. Interestingly, our results corroborate well with previous results [115] and ascertain that yeast Tpm1 provides very little protection from cofilin-mediated severing, compared to the much superior protection provided by the mammalian Tpm1.7. In conjunction

with previous studies [93,115], our results also suggest that *S. cerevisiae* Tpm isoforms may have evolved to allow for faster turnover of actin filaments to support normal rate of actin cable dynamics. Further, our results also present a first comparison of the protective capabilities of Tpm1 and Tpm2 *in vitro*. The fact that Tpm2 can restore not just actin cables but also cable-dependent functions such as vesicle transport and mitochondrial morphology suggest that Tpm1-bound and Tpm2-bound filaments interact with actin-binding proteins like type-V myosin and outer mitochondrial proteins in a similar way (Figs 6A-D and 7B). However, their distinct regulation of Retrograde Actin cable flow (RACF) which is controlled by type-II myosin (Myo1) suggests distinct effects on type-II myosin binding possibly through different overlap with type-II myosin binding site on the actin filament [39]. Structural data about their binding will reveal more insights into these possibilities. From our results, it's clear that Tpm2 is present at limiting levels and a near doubling in its protein amount in cells can restore normal growth and actin cytoskeleton in Δ*tpm1* cells (Figs 5A-C, S7 and 7B). The reason why cells would keep Tpm2 at such low levels are still not clear and one possibility might be to maintain its specific function in RACF-mediated asymmetric inheritance of damaged organelles and protein aggregates between mother and bud [39,40,91,93], while also acting as a latent reservoir for actin cable maintenance in stressed conditions. Consistent with this, we found cells lacking Tpm2 show decreased cable stability in presence of the actin depolymerizing drug LatB demonstrating the contribution of Tpm2 to actin cable stability under conditions of stress (Fig 5D and 5E). Our data suggests that such functional redundancy among diverged Tpm isoforms may confer robustness to actin-dependent processes and increase chances of cell survival under certain stressed conditions. In the future, it remains to be understood whether structural differences in F-actin binding modes [34] or other factors such as post-translational modifications [25,48] in conjunction with expression levels and spatio-temporal regulation, etc. contribute to the shared and distinct roles of closely related Tpm isoforms in *S. cerevisiae* and other eukaryotes.

### Tropomyosin levels control global actin regulation and cross-talk of actin networks

Actin monomeric pool in the cell is used to maintain diverse actin filament network types that differ in geometry, composition of actin-binding proteins, turnover and thus, their function [97]. Tropomyosin majorly binds to linear actin filaments nucleated by formin proteins and prevents branching of linear actin filaments by inhibiting Arp2/3 complex-mediated branch nucleation on existing filaments [116,117]. Linear actin cables and branched actin patches are the two major actin networks in yeast with Tpm majorly localizing to the cables. Our data suggest that global decrease in Tpm levels not only cause a loss of actin cables but also affect actin accumulation and dynamics at actin patches which are sites of endocytosis (Fig 6E-G). This contrasts with observations in *S. pombe* where endocytosis lifetime was similar to wildtype in the presence of a temperature sensitive allele of *cdc8* [96], highlighting differences in actin network regulation between these yeast species despite many similarities. This demonstrates that Tpm1 and Tpm2 levels are important for global actin network homeostasis and suggests that dysregulation of actin cables caused by decreased Tpm levels trigger dysregulation of other actin networks which may contribute to cellular defects observed in Δ*tpm1* cells (Fig 7B). It remains to be addressed whether these effects are entirely due to increased actin accumulation or also accompany changes in ABP composition of the branched actin network. Future work with emphasis on network composition is required to delineate exact molecular basis of such effects in cells with loss of normal actin network homeostasis. Our results, thus, open exciting avenues to understand effects of ABPs on actin networks at a systems level in the future.

## Materials and methods

### Plasmid construction

All plasmids used in this study are listed in S1 Table. Plasmids were constructed by assembling DNA fragments amplified by PCR into restriction endonuclease-digested vectors using NEB Hifi Builder (NewEngland BioLabs; cat.no: E2621L). The plasmids were confirmed with restriction digestion and sequencing. For referencing of plasmids in the text and

**A**

**Spatial sorting of Tpm isoforms**

Tpm1 and Tpm2 co-localize on formin-made actin cables in *S. cerevisiae*

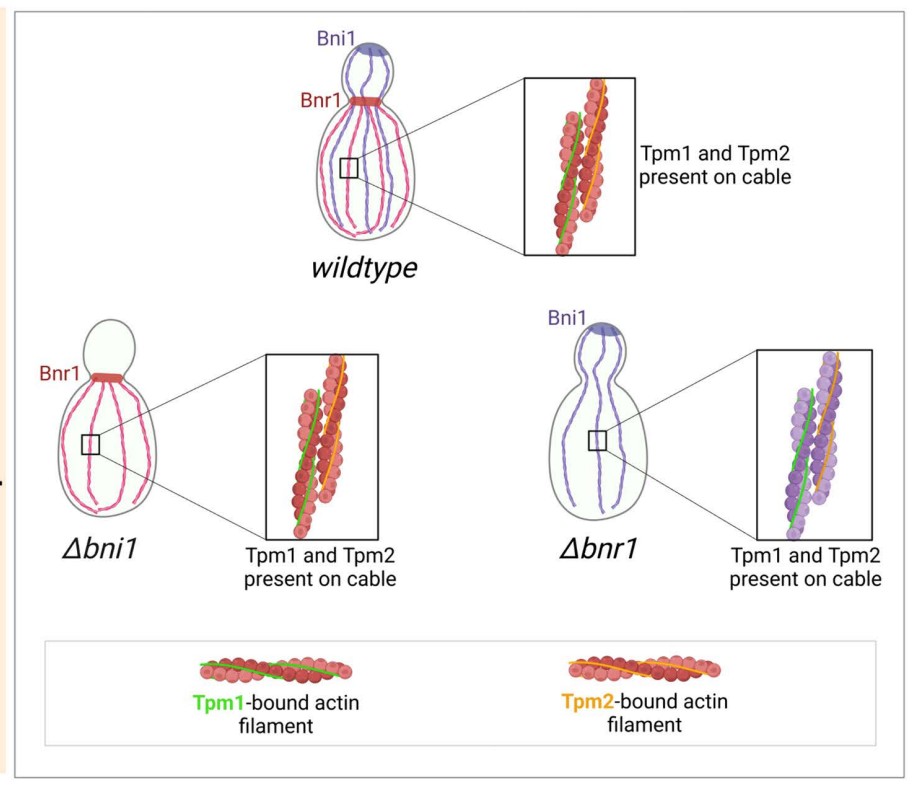

**B**

**Functional redundancy of Tpm isoforms**

Tpm2 can organize a functional actin cable cytoskeleton independent of Tpm1

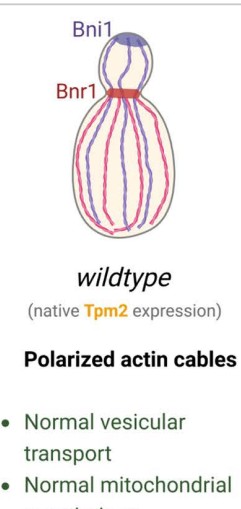 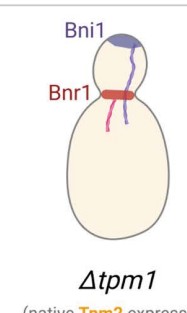 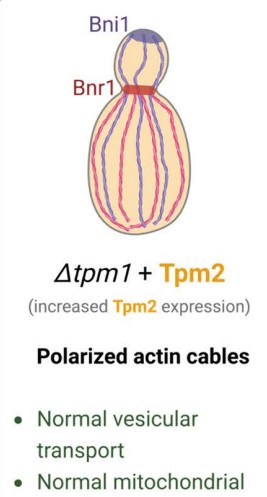

**Fig 7. Schematic Model summarizing: (A) Tpm1 and Tpm2 indiscriminately bind to actin filaments made by formin isoforms Bnr1 and Bni1 *in vivo*. (B)** Tpm2 can independently organize a functional actin cable cytoskeleton in the absence of Tpm1 upon increased expression. Models created using BioRender: https://BioRender.com/suqdvq2; https://BioRender.com/z08hv3u.

figures, we have implemented a consistent and simplified naming system throughout the manuscript, including all figures. In this system, we append a subscripted letter in parentheses to the plasmid backbone name to indicate the plasmid's copy number: (L) for low-copy plasmids (e.g. *pRS316*), (H) for high-copy (e.g. *pRS425*), and (I) for integration (e.g. *pRS305*). For example, a low-copy centromeric plasmid expressing Tpm1 is referred to as *pRS316$_{(L)}$-Tpm1* and a high-copy centromeric plasmid expressing Tpm1 is now referred to as *pRS425$_{(H)}$-Tpm1*. The native Tpm1 or Tpm2 promoters are used for expression unless otherwise mentioned in the text or figures.

## Yeast strains construction

All yeast strains used in this study are listed in S2 Table. S288c genetic background was used for all modifications. Yeast transformation was done using the Lithium Acetate based method as previously described [118]. Deletion and tagging of genes was done using the homologous recombination of PCR cassettes as described previously [119]. The strains were confirmed using colony PCR and fluorescence microscopy.

## Yeast spot assay and growth curve

Yeast cells were grown overnight at 25°C or 30°C. Next day, OD$_{600}$ was normalized to 0.025 or 1 for growth curve and spot assay respectively. For spot assay, 5µL of the normalized culture was spotted on a appropriate plate. For growth curve, 150µL of the culture was seeded into a 96 well plate and allowed to grow till saturation. The OD$_{600}$ was measured every 1 hour during the growth curve experiment. The growth curve data was analyzed in GraphPad Prism to calculate Generation Time.

## Live-cell Imaging of tropomyosin isoforms

Yeast strains were grown overnight in Synthetic Complete (SC) Media at 25°C. The overnight culture was used to inoculate a secondary culture which was allowed to grow till mid-log phase. The cells were then adhered on a Concanavalin A-coated glass-bottom dish (Cellvis cat: no: D35C4-20-1.5-N) containing SC media. Images were acquired using the Andor Dragonfly 502 spinning disk confocal system (Oxford Instruments) equipped with a fully-motorized Leica Dmi8 inverted microscope setup. Z-stacks were acquired using a 100x oil objective, captured with the Andor Sona scMOS camera and deconvolved using Andor Fusion software. The images were processed using Fiji and quantified using a previously described protocol [57,120]. For Signal-to-Noise ratio (SNR) of mNG-Tpm fluorescence on cables, SNR was calculated using the following formula [121]:

$$\text{SNR} = (\text{Max. mNG signal} - \text{Mean background signal}) \div \text{Std. Deviation of background signal}$$

## Actin cable staining with phalloidin

In order to visualize the actin cables, cells were stained with Alexa488/Rhodamine Phalloidin and imaged using a previously described protocol [57,120]. Briefly, cells were grown at 23°C until early-mid log phase in Yeast extract/Peptone/2%-Glucose (YPD) media. The cells were then fixed twice with 4% paraformaldehyde in YPD and 4% paraformaldehyde in 1x PBS for 45 minutes each and washed thrice with 1x PBS. Rhodamine/Alexa488 phalloidin (Invitrogen cat: no: R415 and cat: no: A12379 respectively) was added to a final concentration of 0.4µM and incubated overnight with rotation at 4°C. Next day, the cells were washed thrice with 1X PBS. The cells were adhered on 6% Concanavalin A (Sigma cat: no: C2010) coated glass bottom dishes and z-stacks (step size=o.2µm, 31 slices) were acquired using Andor Dragonfly 502 Spinning Disk system and deconvolved using Andor Fusion Software.

## Actin and Tropomyosin Cable quantification

Actin/Tropomyosin cable length and number was done exactly as previously described [120]. The sum of all cables annotated for length was counted as actin cable number. An example of the quantification procedure and workflow are shown in S10A Fig. For colocalization analysis of ymScarleti-$^{AS}$Tpm1 and mNG-$^{AS}$Tpm2, JACOP plug-in in Fiji was used. The

Pearson's Correlation Coefficient was measured for cells with both channels in the original orientation and with one of the channels flipped horizontally or vertically as a control for comparison.

### Yeast mating experiment

Yeast strains of opposite mating types were grown overnight at 25°C and mixed in equal ratio. The mixture of the two strains was seeded on a 1.2% Agarose pad or on a 6% Concanavalin-A coated glass bottom dish and allowed to grow at 25°C for 30–45 minutes. Time-lapse images were acquired at an interval of 2 min to capture mating events. The time point where cytoplasmic fluorescence was observed to appear in the other mating yeast cell which originally did not have any fluorescence was set as t = 0 (fusion completed).

### Purification and labeling of actin

Rabbit skeletal muscle actin was purified from acetone powder generated from frozen ground hind leg muscle tissue of young rabbits (PelFreez, USA). Lyophilized acetone powder stored at −80°C was mechanically sheared in a coffee grinder, resuspended in G-buffer (5 mM Tris-HCl pH 7.5, 0.5 mM Dithiothreitol (DTT), 0.2 mM ATP, and 0.1 mM $CaCl_2$), and cleared by centrifugation for 20 min at 50,000 × g. The supernatant was collected and further filtered with the Whatman paper. Actin was then polymerized overnight at 4°C, slowly stirring, by the addition of 2 mM $MgCl_2$ and 50 mM NaCl to the filtrate. Next morning, NaCl powder was added to a final concentration of 0.6 M, and stirring was continued for another 30 min at 4°C. F-actin was pelleted by centrifugation for 2.5 hrs at 280,000 × g. The pellet was solubilized by dounce homogenization and dialyzed against G-buffer for 48 h at 4°C. Monomeric actin was precleared at 435,000 × g and loaded onto a Sephacryl S-200 16/60 gel-filtration column (Cytiva, USA) equilibrated in G-Buffer. Fractions containing actin were stored at 4°C.

To biotinylate actin, purified G-actin was first dialyzed overnight at 4°C against G-buffer lacking DTT. The monomeric actin was then polymerized by the addition of an equal volume of 2X labeling buffer (50 mM imidazole pH 7.5, 200 mM KCl, 0.3 mM ATP, 4 mM $MgCl_2$). F-actin was then mixed with a fivefold molar excess of NHS-XX-Biotin (Merck KGaA, Germany) and incubated in the dark for 15 h at 4°C. F-actin was pelleted by centrifugation at 450,000 × g for 40 min at room temperature, The pellet was rinsed with G-buffer, homogenized with a dounce and dialyzed against G-buffer for 48 h at 4°C. Biotinylated monomeric actin was purified further on a Sephacryl S-200 16/60 gel-filtration column as above. Aliquots of biotin-actin were snap-frozen in liquid $N_2$ and stored at −80°C.

To fluorescently label actin, G-actin was polymerized by dialyzing overnight against modified F-buffer (20 mM PIPES pH 6.9, 0.2 mM $CaCl_2$, 0.2 mM ATP, and 100 mM KCl). F-actin was incubated for 2 h at room temperature with a fivefold molar excess of Alexa-488 NHS ester dye (Thermo Fisher Scientific, USA, cat: no: A20100). F-actin was pelleted by centrifugation at 450,000 × g for 40 min at room temperature, and the pellet was resuspended in G-buffer, homogenized with a dounce and further incubated on ice for 2 h to depolymerize the filaments. The monomeric actin was then re-polymerized on ice for 1 h by the addition of 100 mM KCl and 1 mM $MgCl_2$. F-actin was once again pelleted by centrifugation for 40 min at 450,000 × g at 4°C. The pellet was homogenized with a dounce and dialyzed overnight at 4°C against 1 L of G-buffer. Next morning, the solution was precleared by centrifugation at 450,000 × g for 40 min at 4°C. The supernatant was collected, and concentration and labeling efficiency of actin were determined. Labelled actin was stored at 4°C.

### Purification of profilin

Human profilin-1 was expressed in *E. coli* strain BL21 (pRare) to log phase in LB broth at 37°C and induced with 1 mM IPTG for 3 h at 37°C. Cells were then harvested by centrifugation at 15,000 × g at 4°C and stored at −80°C. For purification, pellets were thawed and resuspended in 30 mL lysis buffer (50 mM Tris-HCl pH 8, 1 mM DTT, 1 mM PMSF protease inhibitors (0.5 μM each of pepstatin A, antipain, leupeptin, aprotinin, and chymostatin) was added, and the solution was

sonicated on ice by a tip sonicator. The lysate was centrifuged for 45 min at 120,000 × g at 4°C. The supernatant was then passed over 20 ml of Poly-L-proline conjugated beads in a disposable column (Bio-Rad, USA). The beads were first washed at room temperature in wash buffer (10 mM Tris pH 8, 150 mM NaCl, 1 mM EDTA, and 1 mM DTT) and then washed again with two column volumes of 10 mM Tris pH 8, 150 mM NaCl, 1 mM EDTA, 1 mM DTT, and 3 M urea. Protein was then eluted with five column volumes of 10 mM Tris pH 8, 150 mM NaCl, 1 mM EDTA, 1 mM DTT, and 8 M urea. Pooled and concentrated fractions were then dialyzed in 4 L of 2 mM Tris pH 8, 0.2 mM EGTA, 1 mM DTT, and 0.01% NaN3 for 4 h at 4°C. The dialysis buffer was replaced with fresh 4 L buffer, and the dialysis was continued overnight at 4°C. The protein was centrifuged for 45 min at 450,000 × g at 4°C, concentrated, aliquoted, flash-frozen in liquid N2, and stored at −80°C.

## Purification of yeast and human Tpm isoforms

All budding yeast and human tropomyosins were expressed and purified from *E. coli* strain BL21 to log phase in LB broth at 37°C and induced with 1 mM IPTG for 3 h at 37°C, as done previously [122]. Cells were then harvested by centrifugation at 15,000 × g at 4°C and stored at −80°C. For purification, pellets were thawed and resuspended in 30 mL lysis buffer (20 mM Tris-HCl pH 7.5, 0.5 M NaCl, 5 mM MgCl$_2$, 1 mM PMSF, protease inhibitors (0.5 μM each of pepstatin A, antipain, leupeptin, aprotinin, and chymostatin)) was added, and the solution was sonicated on ice by a tip sonicator. The lysate was incubated at 80°C for 10 min in a water bath and at room temperature for another 10 min. The lysate was centrifuged for 30 min at 30,000 x g at 4°C, and the pellet was discarded. The protein was precipitated by adding 0.3M HCl to the supernatant till the pH reached 4.7. The solution was centrifuged for 30 min at 30,000 x g at 4°C. The pellet was resuspended in 25 mL of wash buffer (100 mM Tris-HCl pH 7.5, 0.5 M NaCl, 5 mM MgCl$_2$, 1 mM DTT). The acid precipitation and accompanying centrifugation step were repeated. The pellet was then resuspended in 25 mL of wash buffer and dialyzed in 2 L of dialysis buffer (20 mM HEPES pH 6.8, 50 mM NaCl, 0.5 mM DTT) overnight at 4°C. The dialyzed solution was loaded to a 5 ml HiTrap Q HP column (Cytiva). The protein fractions were eluted with a linear gradient of NaCl (50–600 mM). The fractions were analyzed by SDS-PAGE to determine the ones containing Tpm. They were concentrated and further loaded on a Superose 6 gel-filtration column (Cytiva) pre-equilibrated with 20 mM Tris (pH 7.5), 50 mM KCl, 2 mM MgCl2, 1 mM DTT. Peak fractions were collected, concentrated, aliquoted, and flash-frozen in liquid N2 and stored at −80°C.

## Purification of yeast cofilin

His-tagged yeast Cof1 was expressed in *E. coli* strain BL21 to log phase in LB broth at 37°C and induced with 1 mM IPTG for 3 h at 37°C. Cells were then harvested by centrifugation at 15,000 × g at 4°C and stored at −80°C. For purification, pellets were thawed and resuspended in 30 mL lysis buffer (50 mM phosphate buffer pH 8, 20 mM Imidazole, 0.3 M NaCl, 1 mM PMSF, protease inhibitors) was added, and the solution was sonicated on ice by a tip sonicator. The lysate was centrifuged for 45 min at 120,000 × g at 4°C. The supernatant was incubated with 2 mL of washed Ni-NTA beads for 2 hrs at 4°C. The solution was centrifuged at 1000 xg for 5 minutes. The supernatant was removed. The beads were washed 3 times with 10 mL of wash buffer (50 mM Phosphate buffer pH 8, 20 mM Imidazole, 0.3 M NaCl, 1 mM DTT) and subsequently centrifuged at 1000 xg for 2 min each time to remove the supernatant. Protein was then eluted with 1 mL of elution buffer (50 mM phosphate buffer pH 8, 250 mM Imidazole, 0.3 M NaCl, 1 mM DTT). Pooled and concentrated fractions were then loaded on a Superose 6 gel-filtration column (Cytiva) pre-equilibrated with 20 mM HEPES pH 7.5, 50 mM KCl, 0.5 mM DTT. Peak fractions were collected, concentrated, aliquoted, and flash-frozen in liquid N2 and stored at −80°C.

## TIRF microscopy

Glass coverslips (60 × 24 mm; Thermo Fisher Scientific, USA) were first cleaned by sonication in detergent for 20 min, followed by successive sonications in 1 M KOH, 1 M HCl, and ethanol for 20 min each. Coverslips were then washed

extensively with $H_2O$ and dried in an $N_2$ stream. The cleaned coverslips were coated with 2 mg/mL methoxy-polyethylene glycol (mPEG)-silane MW 2000 and 2 µg/mL biotin-PEG-silane MW 3400 (Laysan Bio, USA) in 80% ethanol (pH 2.0) and incubated overnight at 70°C. Flow cells were assembled by rinsing PEG-coated coverslips with water, drying with $N_2$, and adhering to µ-Slide VI0.1 (0.1 mm × 17 mm × 1 mm) flow chambers (Ibidi, Germany) with double-sided tape (2.5 cm × 2 mm × 120 µm) and epoxy resin for 5 min (Devcon, USA). Before each reaction, the flow cell was sequentially incubated for 1 min each with 4 µg/ml streptavidin and 1% BSA in 20 mM HEPES pH 7.5 and 50 mM KCl. The flow cell was then equilibrated with TIRF buffer (10 mM imidazole, pH 7.4, 50 mM KCl, 1 mM $MgCl_2$, 1 mM EGTA, 0.2 mM ATP, 10 mM DTT, 2 mM DABCO, and 0.5% methylcellulose [4000 cP]).

Actin filaments were assembled by introducing 1 µM 15% Alexa 488 labelled 0.1% biotinylated actin and 2 µM profilin in the flow cell, with or without 10 µM Tpm. Filaments were allowed to grow for 2–3 min. For experiments containing Tpm, the flow cell was then rinsed with TIRF buffer supplemented with 10 µM Tpm to remove free actin. The solution was then replaced with 10 µM Tpm and 150 nM Cof1. For control experiments containing no Tpm, both washing and cofilin addition step had no Tpm in the solution.

## Image acquisition and analysis

Single-wavelength time-lapse TIRF imaging was performed on a Nikon-Ti2000 inverted microscope equipped with a 40 mW Argon laser, a 60X TIRF-objective with a numerical aperture of 1.49 (Nikon Instruments Inc., USA), and an IXON LIFE 888 EMCCD camera (Andor Ixon, UK). One pixel was equivalent to 144 × 144 nm. Focus was maintained by the Perfect Focus system (Nikon Instruments Inc., Japan). Time-lapse images were acquired every 2 s using Nikon Elements imaging software (Nikon Instruments Inc., Japan). The sample was excited by a 488 nm laser for imaging.

Time-lapse images were analyzed using Fiji [124]. The actin filament severing rates (Fig 4B) were determined by first counting the number of severing events as a function of time on all the filaments in a single field of view. The number of breaks per minute were then divide by total length of all filaments in the field of view to get normalized severing rates. The average length of the filaments post-severing (Fig 4C) was measured following 2 min of exposure to cofilin (with or without Tpm). Data analysis was carried out in Microcal Origin.

## RNA extraction and RT-qPCR

Hot phenol method was used to extract RNA from yeast cells [123]. Overnight grown yeast cells were diluted and allowed to grow until OD at 600nm reached 1. Cells were treated with equal volumes of TES and water saturated acidic phenol and precipitated at -80°C to extract RNA. The extracted RNA was then treated with DNAse A and this was used as a template for first strand cDNA synthesis. Following this, RT-qPCR was set up using the cDNA to check the expression levels. Gene specific primers were used to assess the Tpm1 and Tpm2 levels in wild type and different mutant background. TDH1 and PGK1 housekeeping genes were used as controls for all RT-qPCR experiments.

## Vesicle delivery to bud

Vesicle delivery was visualized using an N-terminal tagged ymScarleti-Sec4 construct expressed under ADH promoter as an additional copy integrated into the *ura3* locus. Overnight grown yeast cells were diluted and allowed to grow until mid-log phase. Cells were adhered on a Concanavalin A-coated glass bottom dish and z-stacks were acquired with a step size of 0.5µm to cover a range of 6µm. The ratio of ymScarleti-Sec4 intensity in the bud and mother were taken and plotted to assess targeting of vesicles to the bud via the actin cable network. Images were also acquired using Andor BC43 table-top spinning disk confocal system using a 60x oil objective and scMOS camera detection. Images were deconvolved using Andor Fusion software and any adjustments for representation were done using Fiji (ImageJ) [124].

## Mitochondrial morphology

Mitochondrial morphology was assessed with Om45–3xmCherry tagged at its genomic locus or Su9-mNeonGreen expressed from an integrated plasmid as a mitochondrial marker during imaging. Z-stacks of mid-log phase cells were acquired using Olympus IX83 widefield fluorescence microscope system and images were processed using the Mitochondria Analyzer plug-in (https://github.com/AhsenChaudhry/Mitochondria-Analyzer) in Fiji [124] to quantify mitochondrial morphology. An example of the quantification workflow is shown in S10B Fig.

## Actin patch imaging

Actin patch lifetime was measured using LifeAct-eGFP as an actin patch marker. Time-lapse images of LifeAct-eGFP expressing cells were acquired at an interval of 0.5 sec for 120 cycles with a 100x oil objective using Olympus SpinSR spinning disk confocal system. Patch lifetime was calculated as the time between appearance and disappearance of LifeAct-eGFP patch fluorescence signal.

## Retrograde Actin Cable Flow

Retrograde actin cable flow was assessed using LifeAct-eGFP as an actin cable marker. Time-lapse images of LifeAct-eGFP expressing cells were acquired at an interval of 1 sec for 20 cycles with a 100x oil objective using Olympus SpinSR spinning disk confocal system. Images were analyzed to measure rate of actin cable elongation towards the rear of the mother cell.

## Latrunculin B sensitivity assay

Yeast cells were grown to mid-log phase at 25°C and LatB (SigmaAldrich, cat. No.: 428020) was added to a final concentration of 67µM. Cells were fixed with 4% paraformaldehyde at time points 0 min, 2 min and 04.30 min after LatB addition. The cells were then stained with Alexa488-phalloidin and imaged as described above.

## Image analysis and statistical analysis

All images were processed using Fiji (ImageJ) [124]. The specific analysis pipelines are mentioned in the above sections for each type of measurement. The graphs were plotted using GraphPad Prism (v.6.04) and statistical tests were also performed using in-built functions in GraphPad Prism. * $p < 0.05$, ** $p < 0.01$, *** $p < 0.001$, ****$p < 0.0001$

## Supporting information

**S1 Fig: Functional characterization of mNeonGreen-Tpm fusion proteins. (A)** Representative time-lapse montages of wildtype yeast cells expressing mNG-Tpm1; scale bar - 3µm. **(B)** Representative time-lapse montages of wildtype yeast cells expressing mNG-Tpm2; scale bar - 3µm. **(C)** Spot assay images for indicated yeast strains performed at 23°C, 30°C, and 37°C. **(D)** Representative images of cells of indicated yeast strains stained with Rhodamine-phalloidin; scale bar – 2µm. **(E)** Plot representing growth curves of indicated yeast strains performed at 37°C. y-axis represents mean absorbance at 600nm. **(F)** Table showing descriptive statistics of Generation Time derived from analysis of growth curve experiment represented in **(E)**. **(G-H)** Representative images of cells expressing indicated mNeonGreen-Tpm fusion proteins stained with Rhodamine-phalloidin; scale bar - 2µm.
(TIF)

**S2 Fig. mNG-Tpm fusion proteins rescue synthetic lethality of cells lacking both endogenous Tpm1 and Tpm2. (A)** Representative time-lapse images acquired at identical settings showing localization of indicated mNG-Tpm variants in wildtype cells. **(B)** Plot depicting Signal-to-Noise Ratio of mNG-Tpm fluorescence on cables from images shown in **(A)**;

n ≥ 42 cables per strain. **(C-D)** Spot assay images for indicated yeast strains performed at 23°C, 30°C, and 37°C to test for rescue of synthetic lethality of Tpm1 and Tpm2.
(TIF)

**S3 Fig. Tpm1 and Tpm2 bind to actin filaments made by formins Bnr1 and Bni1 indiscriminately. (A)** Representative images of cells of wildtype, Δ*bnr1*, and Δ*bni1* cells expressing mNG-^ASTpm1 stained with Rhodamine-phalloidin; scale bar - 2μm. **(B)** Superplot representing mean actin cable length per cell in wildtype, Δ*bnr1*, and Δ*bni1* cells expressing mNG-^ASTpm1; n = 100 cells per strain per replicate, N = 3. **(C)** Superplot representing mean actin cable number per cell in wildtype, Δ*bnr1*, and Δ*bni1* cells expressing mNG-^ASTpm1; n = 100 per strain per replicate, N = 3. **(D)** Representative images of cells of wildtype, Δ*bnr1*, and Δ*bni1* cells expressing mNG-^ASTpm2 stained with Rhodamine-phalloidin; scale bar – 2μm. **(E)** Superplot representing mean actin cable length per cell in wildtype, Δ*bnr1*, and Δ*bni1* cells expressing mNG-^ASTpm2; n = 100 cells per strain per replicate, N = 3. **(F)** Superplot representing mean actin cable number per cell in wildtype, Δ*bnr1*, and Δ*bni1* cells expressing mNG-^ASTpm2; n = 100 cells per strain per replicate, N = 3. **(G)** Representative images depicting localization of mNG-^ASTpm fusion proteins to actin cables in both and bud and mother cell compartment in wildtype cells. **(H)** Box and whiskers plot showing ratio of mean mNG fluorescence in the bud to the mother compartment in the indicated strains; n = 50 cells per strain. Box represents 25th and 75th percentile, line represents median, whiskers represent minimum and maximum value. (Superplots represent datapoints and means from three independent biological replicates marked in different colours; Kruskal-Wallis test with Dunn's Multiple Comparisons test was used in **(B)**, **(C)**, **(E)**, **(F)**; * p < 0.05, ** p < 0.01, *** p < 0.001, **** p < 0.0001)
(TIF)

**S4 Fig. mNG-^ASTpm1 shows binding to both Bnr1- and Bni1-made cables even in the absence of endogenous Tpm1. (A)** Representative images of Δ*tpm1* cells expressing mNG-^ASTpm1 fusion protein as sole copy with indicated genotypes; scale bar - 2μm. **(B)** Superplot representing mean Tpm-bound cable length per cell in indicated strains expressing mNG-^ASTpm1 as sole copy; n > 15 cells per strain per replicate, N = 3. **(C)** Superplot representing mean Tpm-bound cable number per cell in indicated strains expressing mNG-^ASTpm1 as a sole copy; n ≥ 15 cells per strain per replicate, N = 3. **(D)** Representative images of Δ*tpm1* cells expressing mNG-^ASTpm1 fusion protein as sole copy with indicated genotypes stained with Rhodamine-phalloidin; scale bar - 2μm. **(E)** Superplot representing mean actin cable length per cell in indicated yeast strains shown in **(D)**; n = 15 cells per strain per replicate, N = 3. **(F)** Superplot representing mean actin cable number per cell in in indicated yeast strains shown in **(D)**; n = 15 per strain per replicate, N = 3. (Superplots represent datapoints and means from three independent biological replicates marked in different colours; Kruskal-Wallis test with Dunn's Multiple Comparisons test was used in **(B)**, **(C)**, **(E)**, **(F)**; * p < 0.05, ** p < 0.01, *** p < 0.001, **** p < 0.0001)
(TIF)

**S5 Fig. Tpm2 overexpression restores actin cables in Δ*tpm1* cells. (A)** Spot assay image for indicated yeast strains performed at 23°C, 30°C, and 37°C. **(B)** Representative images of indicated yeast strains stained with Alexa488-phalloidin; scale bar – 2μm. **(C)** Plot representing mean percentage of cells with detectable actin cables in indicated yeast strains averaged over 3 biological replicates; n ≥ 200 cells for each strain per replicate, N = 3. **(D)** Plot representing mean actin cable length per cell in the indicated yeast strains as per experiment in **(A)**; n = 30 cells per strain. **(E)** Plot representing actin cable number per cell in the indicated yeast strains as per experiment in **(A)**; n = 30 cells for each strain. **(F)** Plot representing fold change of Tpm2 transcript levels normalized to wildtype in the indicated yeast strains containing high-copy number plasmids; n = 3 per strain per experiment, N = 3. (Box represents 25th and 75th percentile, line represents median, whiskers represent minimum and maximum value; One-Way Anova with Tukey's Multiple Comparisons test was used in **(C)**, **(D)** and **(E)**. * p < 0.05, ** p < 0.01, *** p < 0.001, **** p < 0.0001)
(TIF)

**S6 Fig. GAL1 promoter driven (Galactose-induced) expression of Tpm2 restores actin cables in Δ*tpm1* cells.**
**(A)** Spot assay image for indicated yeast strains performed at 23°C, 30°C, and 37°C in the presence of 0%, 2%, and 4% galactose. **(B)** Representative images of cells with indicated genotypes grown in 4% galactose overnight and stained with Alexa488-Phalloidin.
(TIF)

**S7 Fig. Low-copy plasmid expression of Tpm2 rescues growth and actin cable defects in Δ*tpm1* cells. (A)** Spot assay image for indicated yeast strains performed at 23°C, 30°C, and 37°C. **(B)** Plot representing actin cable number per cell in the indicated yeast strains as per experiment in Fig 5A; n = 50 cells per strain. **(C)** Plot representing fold change of Tpm2 transcript levels normalized to *wildtype* in the indicated yeast strains containing low-copy number centromeric plasmids; n = 3 per strain per experiment, N = 3.
(TIF)

**S8 Fig. Increased protein levels of mNG-$^{AS}$Tpm2 cause a rescue of phenotype in Δ*tpm1* cells. (A)** Spot Assay showing growth of indicated yeast strains at 23°C, 30°C, and 37°C. **(B)** Representative images of yeast cells with indicated genotypes, scale bar - 2μm. **(C)** Plot showing mean mNG fluorescence intensity per cell in the indicated yeast strains, n > 174 cells per strain. **(D)** Representative image of western blot probed with anti-mNG and anti-actin (loading control). **(E)** Plot showing normalized fold change of mNG/actin signal intensity for blot shown in **(D)**, N = 3. (One-Way Anova with Tukey's Multiple Comparisons test was used in **(C)** and **(E)**. * p < 0.05, ** p < 0.01, *** p < 0.001, **** p < 0.0001)
(TIF)

**S9 Fig. Tpm2 overexpression restores actin cable dependent functions in Δ*tpm1* cells. (A)** Representative images of indicated yeast strains expressing ymScarleti-Sec4; scale bar – 2μm. **(B)** Representative images of indicated yeast strains expressing Su9-mNeonGreen; scale bar – 2μm. **(C)** Plot representing ratio of bud/mother ymScarleti-Sec4 fluorescence per cell in the indicated yeast strains; n ≥ 24 cells per strain. **(D)** Plot representing mean mitochondrial fragment count per cell in the indicated yeast strains; n ≥ 25 cells per strain. **(E)** Representative time-lapse montages of wildtype yeast cells expressing LifeAct-eGFP from its native locus, scale bar - 2μm. **(F)** Plot representing retrograde actin cable flow rate (μm/s) in the indicated yeast strains; n ≥ 20 events per strain. (Box represents 25$^{th}$ and 75$^{th}$ percentile, line represents median, whiskers represent minimum and maximum value; One-Way Anova with Tukey's Multiple Comparisons test was used in **(C)**, **(F)**; Kruskal-Wallis test with Dunn's multiple comparisons was used in **(D)**; * p < 0.05, ** p < 0.01, *** p < 0.001, **** p < 0.0001)
(TIF)

**S10 Fig. Analysis pipelines used for quantitative analysis of actin cables and mitochondrial morphology in our study. (A)** Schematic showing quantification workflow for actin cables as adapted from McInally et al. 2022 (*bio-protocol*) [120]. **(B)** Schematic showing quantification workflow for mitochondrial morphology analysis using Mitochondria Analyzer Plug-In (https://github.com/AhsenChaudhry/Mitochondria-Analyzer) in Fiji [124].
(TIF)

**S1 Movie. Time-lapse imaging of small-, medium- and large-budded cells expressing mNeonGreen-$^{AS}$Tpm1. (Scale bar - 3μm).**
(AVI)

**S2 Movie. Time-lapse imaging of small-, medium- and large-budded cells expressing mNeonGreen-$^{AS}$Tpm2. (Scale bar - 3μm).**
(AVI)

**S3 Movie.** Time-lapse imaging of haploid yeast cell expressing mNeonGreen-[AS]Tpm1 mating with another wildtype haploid cell. (Scale bar - 3μm).
(AVI)

**S4 Movie.** Time-lapse imaging of haploid yeast cell expressing mNeonGreen-[AS]Tpm2 mating with another wildtype haploid cell. (Scale bar - 3μm).
(AVI)

**S1 Table.** List of plasmids used in this study.
(DOCX)

**S2 Table.** List of yeast strains used in this study.
(DOCX)

**S3 Table.** Graphed values for all the graphs represented in the main figures.
(XLSX)

**S4 Table.** Graphed values for all the graphs represented in the supplementary figures.
(XLSX)

## Acknowledgments

We thank the Department of Biochemistry, Indian Institute of Science for access to DST-FIST imaging and other central facilities. We also thank the Divisional Bioimaging Facility. We are grateful to Mr. Pabitra Sharma for help with image analysis. We thank Prof. PN Rangarajan, Prof. Ramanujam Srinivasan, Prof. Sunil Laxman, Prof. Sachin Kotak, Prof. Sunish Radhakrishnan, and Prof. Marko Kaksonen for their feedback on the manuscript. AD and JK acknowledge GATE fellowship from IISc. JSB acknowledges KVPY fellowship from IISc. We thank Dr. Silvia Jansen for advice on quantification of TIRF microscopy experiments and Blake Miller for help with protein purification.

## Author contributions

**Conceptualization:** Anubhav Dhar, Saravanan Palani.

**Data curation:** Anubhav Dhar, VT Bagyashree, Sudipta Biswas, Jayanti Kumari, Amruta Sridhara, Jeevan Subodh B.

**Formal analysis:** Anubhav Dhar, VT Bagyashree, Sudipta Biswas, Jayanti Kumari, Amruta Sridhara, Jeevan Subodh B, Shashank Shekhar, Saravanan Palani.

**Funding acquisition:** Shashank Shekhar, Saravanan Palani.

**Investigation:** Anubhav Dhar, VT Bagyashree, Sudipta Biswas, Jayanti Kumari, Shashank Shekhar, Saravanan Palani.

**Methodology:** Anubhav Dhar, VT Bagyashree, Sudipta Biswas, Jayanti Kumari.

**Project administration:** Saravanan Palani.

**Resources:** Anubhav Dhar, VT Bagyashree, Jayanti Kumari, Saravanan Palani.

**Supervision:** Shashank Shekhar, Saravanan Palani.

**Validation:** Anubhav Dhar, VT Bagyashree, Sudipta Biswas, Jayanti Kumari, Amruta Sridhara, Jeevan Subodh B, Shashank Shekhar.

**Visualization:** Anubhav Dhar, VT Bagyashree, Sudipta Biswas, Jayanti Kumari, Amruta Sridhara, Jeevan Subodh B.

**Writing – original draft:** Anubhav Dhar, VT Bagyashree, Saravanan Palani.

**Writing – review & editing:** Anubhav Dhar, VT Bagyashree, Sudipta Biswas, Jayanti Kumari, Amruta Sridhara, Shashank Shekhar, Saravanan Palani.

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
