## [Decision Letter · Decision Letter 0]

11 Jun 2025

PGENETICS-D-25-00504

Functional redundancy and formin-independent localization of tropomyosin isoforms in Saccharomyces cerevisiae

PLOS Genetics

Dear Saravanan,

Thank you for submitting this interesting and exciting manuscript to PLOS Genetics. After careful consideration, we feel that it has merit but does not fully meet PLOS Genetics's publication criteria as it currently stands. Therefore, we invite you to submit a revised version of the manuscript that addresses the points raised during the review process.

The manuscript was reviewed by three reviewers, all of whom are experts in the field.  Two of the reviewers checked the box for minor revisions, and the third reviewer checked the major revision box.  My reading of their reviews is that addressing them will help sharpen the focus and conclusions, and also ensure that the writing is as tight as it can be such that conclusions are fully supported or appropriately nuanced, the figures are referenced appropriately, and the questions raised by review have been addressed and resolved.  The reviewers noted that much of the approach is cell biological, and thus one might see this type of study in MBOC, JCB, MCB.  My own impression is that there is enough of a yeast genetics slant here that PLOS Genetics is just fine from this editor's perspective.  The system automatically provides 60 days for revision. This may be just fine, but if you find you need a bit more time to revise please just let us know.

Please submit your revised manuscript within 60 days Aug 10 2025 11:59PM. If you will need more time than this to complete your revisions, please reply to this message or contact the journal office at plosgenetics@plos.org. Please include the following items when submitting your revised manuscript:

We look forward to receiving your revised manuscript and thank you again for submitting this exciting and interesting work to PLOS Genetics.

Kind regards,

Joseph Heitman, MD, PhD

Academic Editor

PLOS Genetics

Geraldine Butler

Section Editor

PLOS Genetics

Aimée Dudley

Editor-in-Chief

PLOS Genetics

Anne Goriely

Editor-in-Chief

PLOS Genetics

**Journal Requirements:**

https://journals.plos.org/plosgenetics/s/submission-guidelines#loc-parts-of-a-submission

4) We notice that your supplementary Figures, and Tables are included in the manuscript file. Please remove them and upload them with the file type 'Supporting Information'. Please ensure that each Supporting Information file has a legend listed in the manuscript after the references list.

Potential Copyright Issues:

i) Figures 4D, 7A, and 7B. Please confirm whether you drew the images / clip-art within the figure panels by hand. If you did not draw the images, please provide (a) a link to the source of the images or icons and their license / terms of use; or (b) written permission from the copyright holder to publish the images or icons under our CC BY 4.0 license. Alternatively, you may replace the images with open source alternatives. See these open source resources you may use to replace images / clip-art:

ii) Figure S7A appears to have been adapted from a previously published figure. Please provide written permission from the copyright holder to publish this under our CC-BY 4.0 license, or remove the figure / replace the image. Please note we do not recommend using standard request forms available on Publishers' websites, as they grant single use rather than republication under an open access license.

6) We note that your Data Availability Statement is currently as follows: "All relevant data are within the manuscript and its Supporting information files.". Please confirm at this time whether or not your submission contains all raw data required to replicate the results of your study. Authors must share the “minimal data set” for their submission. PLOS defines the minimal data set to consist of the data required to replicate all study findings reported in the article, as well as related metadata and methods (https://journals.plos.org/plosone/s/data-availability#loc-minimal-data-set-definition).

7) Please amend your detailed Financial Disclosure statement. This is published with the article. It must therefore be completed in full sentences and contain the exact wording you wish to be published.

2) If any authors received a salary from any of your funders, please state which authors and which funders..

**Reviewers' comments:**

Reviewer's Responses to Questions

**Comments to the Authors:**

Reviewer #1: The authors present a thorough analysis for how different isoforms of tropomyosin in budding yeast mechanistically maintain and organize actin cables. They utilize novel tagged, yet functinal proteins for both isoforms of tropomyosin for this anaylsis. They found that the different isoforms are able to bind actin cables indepenedent of whic formin nucleated the cable, this finding is distinct from previous reports done in other organisms, showing that there is diversity in actin-binding protein funciton across species. They reveal novel functions for Tpm2, in that it, but not Tpm1 can regulate retrgorade actin cable flow, and that Tpm2 can function as the sinlge copy of tropomyosin, as long as it is expressed in sufficient quantites. They also show that the concentration of tropomyosin in the cell, and not a specific isoform, is responsible to manintaining the balance between actin networks. The manuscript presents strong evidence for their claims and their conclusions match the data presented. This work is a significant step forward for the cytoskeletal field and presents novel tools that can be an important contribution to other labs in this area of research.

Minor Issues:

1) Define actin binding protein as (ABP) in the abstract, as the authors use the abbreviation in the introduction without making the distinction.

2) Move the OM45-mCherry images from Figure S1I to Figure 1 to match with Figure 1D. Having the images and subsequent analysis on different figures is very awkward.

3) Please report n values for charts in Figure 1C and 1D.

4) In Figure 2 and S2, the authors utilize phalloidin staining as a readout with deletion of formins. They state that their tagged tropomyosins were unstable when fixed, thus separate analysis. However, they have ymScarleti-AS-Tpm1 later in the figure and LifeAct-eGFP later in the manuscript, so co-expressing these together would allow for simultaneous analysis without fixation.

5) There is no quantification for Figure 2G with the colocalization of the 2 isoforms, so while they have a single image, quantifcation of the colocalization would strengthen the argument by showing that colocalization is a common occurance.

6) In Figure 4 with the TIRF analysis, the authors utilize yeast cofilin and the yeast tropomyosins, but then use human profilin. Budding yeast profilin is easily purified and functions with actin in TIRF, and while human profilin can partially compensate for a budding yeast profilin mutant, it would be a cleaner experiment using budding yeast profilin, just to eliminate any variables.

7) In the text about Tpm2 mRNA levels on page 8, the authors cite their Figure S4F, when it relates to the data in Figure S4E.

8) In the Discussion section on page 13, the authors talk about the LatB experiment, but then cite Figure 4G and 4H, when the data is in Figure 5D and 5E.

9) In the Discussion section on page 14, the authors talk about the LifeAct-eGFP actin patch experiments, but then cite Figure 5E-G, when the data is in Figure 6E-G.

Reviewer #2: Reviews are uploaded as an attachment.

Reviewer #3: Key Results

This manuscript from Dhar and colleagues addresses the role of two yeast tropomyosin isoforms, Tpm1 and Tpm2, in the regulation of actin cable architecture and function in budding yeast S. cerevisiae. They created novel mNeonGreen-tagged acetylation-mimic (AM) Tropomyosin probes expressed as the sole gene copy at the endogenous loci for both Tpm1 and Tpm2 that complemented cell growth and actin cable architecture in vivo. These new fluorescent probes for visualizing tropomyosin decoration of actin cables allowed the group to show Tpm1 and Tpm2 decorate cables generated by either formin Bni1 or Bnr1. Further, Tpm1 and Tpm2 bind actin cables indiscriminately in cells undergoing mating-induced fusion or miotic growth and division. The group used purified proteins to demonstrate that Tpm1 and Tpm2 equally protected actin filaments from cofilin-mediated severing. They also show that contrary to previous reports, increased expression of Tpm2 complements the cell growth defects and actin cable architecture of tpm1D, demonstrating that Tpm1 and Tpm2 are more functionally interchangeable than previously thought. Interestingly, even mild (2-fold) overexpression of Tpm2 is sufficient to suppress the loss of actin cables in tpm1D. Moreover, they show that actin cables in tpm2D cells are hyper-sensitive to latrunculin, which represents one of the first phenotypes found for tpm2?. Lastly, the group found that Tpm2 overexpression not only restores actin cables in tpm1D cells, but also rescues defects in mitochondrial morphology, secretory vesicle polarization, and endocytic actin patch dynamics. These results conclude in a model showing that Tpm1 and Tpm2 co-occupy actin cables generated by either yeast formin Bni1 or Bnr1 and that elevated Tpm2 levels suppress tpm1D actin-related phenotypes.

Validity

This manuscript utilizes multi-prong approach combining powerful yeast genetics, quantitative live and fixed-cell imaging as well as in vitro reconstitution using purified proteins. This multi-faceted approach to understanding the mechanistic basis for yeast tropomyosin function in vivo is well-rounded. The genetic data in this paper are robust, and conducted in a rigorous manner. The authors use modern quantitative measurements of actin cables (McInally et al., 2022) in their analyses of both phalloidin-stained and Tpm-decorated actin cables.

Significance

This manuscript contains a number of significant findings that will be well received by the cell biology and biochemistry fields. The generation of functional endogenously-tagged tropomyosins in vivo provides a powerful tool for the yeast actin field to dissect Tpm1 localization and function beyond this study. Before this manuscript, a previous paper (Drees et al., 1995) reported that although deletion of TPM1 and TPM2 together was lethal in yeast, overexpression of Tpm2 could not rescue tpm1D. Instead, Dhar and colleagues show that this result was not correct, and that Tpm2 overexpression restores actin cables and other functions in tpm1D. Uncovering how yeast utilize two tropomyosin isoforms will appeal to a broad audience in the actin field, since higher order eukaryotes typically possess multiple tropomyosins.

Data and methodology

The quality and presentation of the cell biological, biochemical and genetic data is sound overall. However, the actin cable quantification methodologies require some additional explanation for readers. For quantification of actin cable length and number, the authors simply cite McInally et al., 2022 and show an example image analysis pipeline. However, McInally et al., 2022 only demonstrates a quantitative analysis pipeline for actin cable length, not cable number. Additionally, the authors claim that fixation disrupts tropomyosin decoration with actin cables and thus forced them to measure Tpm1/Tpm2 coated cables from live cell images as a population average. No methods were provided to explain how they determined Tpm1-coated cable length and number, as this was not provided in McInally et al., 2022.

Analytical approach

The authors provide extensive quantitative analyses wherever appropriate in each figure. However, I do have a concern about graphs in Figure 2B and 2E, in measuring actin cable length differences in various formin mutant backgrounds. By eye, there does not seem to be appreciable differences between the data means and distributions, yet the authors claim bnr1D cells have significantly longer cables. In addition to the graphs not appearing correct by eye, bnr1D cells would be expected to have shorter cables, since they lack the formin responsible for generating long, thick cables in the mother cell compartment. Further explanation or speculation is needed on this.

Suggested Improvements

· In Figure 1A, the authors claim that the acetylation mimic tropomyosin constructs yield clearer and brighter cable staining than the non acetylation mimic constructs shown in Figure S1A. To make this claim, it would be more convincing to show each construct side-by-side with equally contrasted images acquired with identical settings. A fluorescence quantification of cables would also help this be more convincing.

· In Figure 1D, the authors quantify mitochondrial fragmentation as a product of actin cable function. However, the authors fail to properly introduce why this is a viable assay for quantifying cable function. A brief explanation and citing a previous work would be beneficial to the reader. Also, the authors did not call out Figure S1I as the representative mitochondrial images to accompany the graph in Figure 1D.

· For quantifying Tpm-bound cables, the authors revert to using mNG-AS-Tpm constructs expressed as a second copy from a separate locus in the genome while endogenous unlabeled Tpm1 is simultaneously being expressed. This seems odd, since the take-home message from Figure 1 was the functionality of these new endogenously expressed mNG-AS-Tpm constructs. Could the authors provide some rational for this deviation in the text and potentially in the methods section.

· In Figure 2C there is no difference in Tpm-bound cable number between bnr1D and bni1D cells expressing mNG-AS-Tpm1, but in Figure 2F there is a difference in between Tpm-bound cable number between bnr1Dand bni1D cells expressing mNG-AS-Tpm2. The authors do not provide any explanation in the text for this discrepancy.

· In Figure 3, the authors do not describe how they mark the onset of fusion. A more detailed methods section for mating and fusing yeast would be helpful.

· In Figure 3A and Figure 3B, it is very difficult to discern visible Tpm-coated cables in the mating yeast. Clearly, the expression levels decrease as quantified in Figure 3C and Figure 3D, but it remains unclear to me how well Tpm decorates cables in the mating yeast based on the provided representative images.

· There are inconsistencies in data sets between Figure S2 and Figure S3 that are not described in the text:

o In Figure S2, cable length analyses in wildtype, bnr1D and bni1D cells differ depending on whether mNG AS-Tpm1 or mNGAS-Tpm2 are expressed.

o In Figure S3, bni1D cells contain significantly longer actin cables compared to bnr1D cells specifically when using phalloidin staining instead of measuring mNG AS-Tpm1 signal along cables.

o These inconsistencies make it difficult to interpret the role of each formin in regulating actin cable architecture.

· For Figure 4, the authors do not mention that their results (about Tpm1 and Tpm2 protecting actin filaments from cofilin-mediated severing) potentially conflict with a previous study from Fan et al., 2008. This will help clarify things for the field.

· In the text describing Figure S4A-C, the authors detail how their Tpm2 overexpression experiments differ from Drees et al., 1995. The authors claim that the previous work, which expressed Tpm2 on a high-copy plasmid under control of a galactose-inducible promoter, likely generated a lethal dosage of Tpm2, leading to no rescue of tpm1D actin phenotypes. While this is very plausible, given that a mild overexpression of Tpm2 shown in this paper can suppress tpm1D actin phenotypes, it would be significantly more convincing to do a side-by-side comparison using the same plasmid-promoter system Drees et al., 1995 used along with the plasmids used in the current study. I realize that this is asking for more experiments where the answer from the data in the present study is already clear. However, doing them all side-by-side will lay the issue to rest once and for all, and provide final clarity on the genetic relationship between Tpm1 and Tpm2.

Clarity and context

· For Figure 2A and Figure 2D, it would be helpful to use arrows to point to obvious Tpm-coated cables since the Tpm1 constructs have high cytosolic levels.

· Throughout the paper, different plasmids are used to express combinations of Tpm isoforms and promoters. It would be extremely helpful for the readers if the group could create a legend with simplified descriptions of the plasmids properties (i.e. high expression, low expression) and associating these labels when the plasmids are mentioned. Including these simplified labels in the figures would allow the reader to quickly understand what is different between pRS backbones.

**Have all data underlying the figures and results presented in the manuscript been provided?**

Reviewer #1: Yes

Reviewer #2: **No:** Multiple references to the figures are wrong. I'm not sure all the figures are properly called in the text.

Reviewer #3: None

PLOS authors have the option to publish the peer review history of their article (what does this mean? ). If published, this will include your full peer review and any attached files.

**Do you want your identity to be public for this peer review?** For information about this choice, including consent withdrawal, please see our Privacy Policy .

Reviewer #1: No

Reviewer #2: No

Reviewer #3: No

**Figure resubmission:**
---

## [Editor Report · Decision Letter 1]

1 Sep 2025

Dear Dr Palani,

We are pleased to inform you that your revised manuscript entitled "Functional redundancy and formin-isoform independent localization of tropomyosin paralogs in Saccharomyces cerevisiae" has been editorially accepted for publication in PLOS Genetics. Congratulations!

Yours sincerely,

Joe

Joseph Heitman, MD, PhD

Academic Editor

PLOS Genetics

Geraldine Butler

Section Editor

PLOS Genetics

Aimée Dudley

Editor-in-Chief

PLOS Genetics

Anne Goriely

Editor-in-Chief

PLOS Genetics

Comments from the reviewers (if applicable):

**Data Deposition**

http://datadryad.org/submit?journalID=pgenetics&manu=PGENETICS-D-25-00504R1

**Press Queries**

---

## [Editor Report · Acceptance letter]

PGENETICS-D-25-00504R1

Functional redundancy and formin-isoform independent localization of tropomyosin paralogs in Saccharomyces cerevisiae

Dear Dr Palani,

We are pleased to inform you that your manuscript entitled "Functional redundancy and formin-isoform independent localization of tropomyosin paralogs in Saccharomyces cerevisiae" has been formally accepted for publication in PLOS Genetics! Your manuscript is now with our production department and you will be notified of the publication date in due course.

With kind regards,

Anita Estes

PLOS Genetics

On behalf of:
